# Augmented Sliced Wasserstein Distances

**Xiongjie Chen**[1], **Yongxin Yang**[2,3], and **Yunpeng Li**[1]

[1]University of Surrey, [2]University of Edinburgh, [3]Huawei Noah's Ark Lab
[1]{xiongjie.chen, yunpeng.li}@surrey.ac.uk, [2]yongxin.yang@ed.ac.uk

## Abstract

While theoretically appealing, the application of the Wasserstein distance to large-scale machine learning problems has been hampered by its prohibitive computational cost. The sliced Wasserstein distance and its variants improve the computational efficiency through the random projection, yet they suffer from low accuracy if the number of projections is not sufficiently large, because the majority of projections result in trivially small values. In this work, we propose a new family of distance metrics, called augmented sliced Wasserstein distances (ASWDs), constructed by first mapping samples to higher-dimensional hypersurfaces parameterized by neural networks. It is derived from a key observation that (random) linear projections of samples residing on these hypersurfaces would translate to much more flexible *nonlinear* projections in the original sample space, so they can capture complex structures of the data distribution. We show that the hypersurfaces can be optimized by gradient ascent efficiently. We provide the condition under which the ASWD is a valid metric and show that this can be obtained by an injective neural network architecture. Numerical results demonstrate that the ASWD significantly outperforms other Wasserstein variants for both synthetic and real-world problems.

## 1 Introduction

Comparing samples from two probability distributions is a fundamental problem in statistics and machine learning. The optimal transport (OT) theory (Villani, 2008) provides a powerful and flexible theoretical tool to compare degenerative distributions by accounting for the metric in the underlying spaces. The Wasserstein distance, which arises from the optimal transport theory, has become an increasingly popular choice in various machine learning domains ranging from generative models to transfer learning (Gulrajani et al., 2017; Arjovsky et al., 2017; Kolouri et al., 2019b; Cuturi and Doucet, 2014; Courty et al., 2016).

Despite its favorable properties, such as robustness to disjoint supports and numerical stability (Arjovsky et al., 2017), the Wasserstein distance suffers from high computational complexity especially when the sample size is large. Besides, the Wasserstein distance itself is the result of an optimization problem — it is non-trivial to be integrated into an end-to-end training pipeline of deep neural networks, unless one can make the solver for the optimization problem differentiable. Recent advances in computational optimal transport methods focus on alternative OT-based metrics that are computationally efficient and solvable via a differentiable optimizer (Peyré and Cuturi, 2019). Entropy regularization is introduced in the Sinkhorn distance (Cuturi, 2013) and its variants (Altschuler et al., 2017; Dessein et al., 2018) to smooth the optimal transport problem; as a result, iterative matrix scaling algorithms can be applied to provide significantly faster solutions with improved sample complexity (Genevay et al., 2019).

An alternative approach is to approximate the Wasserstein distance through *slicing*, i.e. linearly projecting, the distributions to be compared. The sliced Wasserstein distance (SWD) (Bonneel et al., 2015) is defined as the expected value of Wasserstein distances between one-dimensional random projections of high-dimensional distributions. The SWD shares similar theoretical properties with the Wasserstein distance (Bonnotte, 2013) and is computationally efficient since the Wasserstein distance in one-dimensional space has a closed-form solution based on sorting. (Deshpande et al., 2019) extends the sliced Wasserstein distance to the max-sliced Wasserstein distance (Max-SWD), by finding a single projection direction with the maximal distance between projected samples. The subspace robust Wasserstein distance extends the idea of slicing to projecting distributions on linear subspaces (Paty and

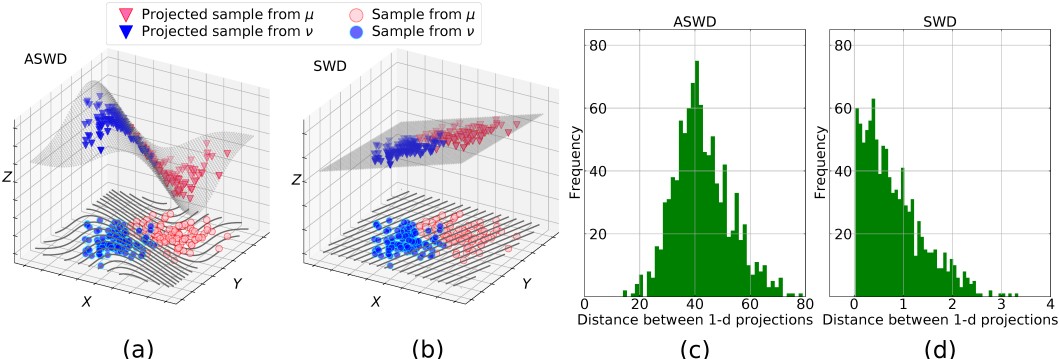

Figure 1: (a) and (b) are visualizations of projections for the ASWD and the SWD between two 2-dimensional Gaussians. (c) and (d) are distance histograms for the ASWD and the SWD between two 100-dimensional Gaussians. Figure 1(a) shows that the injective neural network embedded in the ASWD learns data patterns (in the $X$-$Y$ plane) and produces well-separate projected values ($Z$-axis) between distributions in a random projection direction. The high projection efficiency of the ASWD is evident in Figure 1(c), as almost all random projection directions in a 100-dimensional space lead to significant distances between 1-dimensional projections. In contrast, random linear mappings in the SWD often produce closer 1-d projections ($Z$-axis) (Figure 1(b)); as a result, a large percentage of random projection directions in the 100-d space result in trivially small distances (Figure 1(d)), leading to a low projection efficiency in high-dimensional spaces.

Cuturi, 2019). However, the linear nature of these projections usually leads to low projection efficiency of the resulted metrics in high-dimensional spaces (Deshpande et al., 2019; Kolouri et al., 2019a).

Different variants of the SWD have been proposed to improve the projection efficiency of the SWD, either by introducing nonlinear projections or by optimizing the distribution of random projections. Specifically, (Kolouri et al., 2019a) extends the connection between the sliced Wasserstein distance and the Radon transform (Radon, 1917) to introduce generalized sliced Wasserstein distances (GSWDs) by utilizing generalized Radon transforms (GRTs), which are defined by nonlinear defining functions and lead to nonlinear projections. A variant named the GSWD-NN was proposed in (Kolouri et al., 2019a) to generate nonlinear projections *directly* with neural network outputs, but it does not fit into the theoretical framework of the GSWD and does not guarantee a valid metric. In contrast, the distributional sliced Wasserstein distance (DSWD) and its nonlinear version, the distributional generalized sliced Wasserstein distance (DGSWD), improve their projection efficiency by finding a distribution of projections that maximizes the expected distances over these projections. The GSWD and the DGSWD exhibit higher projection efficiency than the SWD in the experiment evaluation, yet they require the specification of the particular form of defining functions from a limited class of candidates. However, the selection of defining functions is usually a task-dependent problem and requires domain knowledge, and the impact on performance from different defining functions is still unclear.

In this paper, we present the augmented sliced Wasserstein distance (ASWD), a distance metric constructed by first mapping samples to hypersurfaces in an *augmented* space, which enables flexible nonlinear slicing of data distributions for improved projection efficiency (See Figure 1). Our main contributions include: (i) We exploit the capacity of nonlinear projections employed in the ASWD by constructing injective mapping with arbitrary neural networks; (ii) We prove that the ASWD is a valid distance metric; (iii) We provide a mechanism in which the hypersurface where high-dimensional distributions are projected onto can be optimized and show that the optimization of hypersurfaces can help improve the projection efficiency of slice-based Wasserstein distances. Hence, the ASWD is data-adaptive, i.e. the hypersurfaces can be learned from data. This implies one does not need to manually design a function from the limited class of candidates; (iv) We demonstrate superior performance of the ASWD in numerical experiments for both synthetic and real-world datasets.

The remainder of the paper is organized as follows. Section 2 reviews the necessary background. We present the proposed method and its numerical implementation in Section 3. Related work are discussed in Section 4. Numerical experiment results are presented and discussed in Section 5. We conclude the paper in Section 6.

## 2 BACKGROUND

In this section, we provide a brief review of concepts related to the proposed work, including the Wasserstein distance, (generalized) Radon transform and (generalized) sliced Wasserstein distances.

**Wasserstein distance:** Let $P_k(\Omega)$ be a set of Borel probability measures with finite $k$-th moment on a Polish metric space $(\Omega, d)$ (Villani, 2008). Given two probability measures $\mu, \nu \in P_k(\Omega)$, the Wasserstein distance of order $k \in [1, +\infty)$ between $\mu$ and $\nu$ is defined as:

$$W_k(\mu, \nu) = \left( \inf_{\gamma \in \Gamma(\mu, \nu)} \int_{\Omega \times \Omega} d(x, y)^k d\gamma(x, y) \right)^{\frac{1}{k}}, \tag{1}$$

where $d(\cdot, \cdot)^k$ is the cost function, $\Gamma(\mu, \nu)$ represents the set of all transportation plans $\gamma$, i.e. joint distributions whose marginals are $\mu$ and $\nu$, respectively.

While the Wasserstein distance is generally intractable for high-dimensional distributions, there are several favorable cases where the optimal transport problem can be efficiently solved. If $\mu$ and $\nu$ are continuous one-dimensional measures defined on a linear space equipped with the $L^k$ norm, the Wasserstein distance between $\mu$ and $\nu$ has a closed-form solution (Peyré and Cuturi, 2019):

$$W_k(\mu, \nu) = \left( \int_0^1 |F_\mu^{-1}(z) - F_\nu^{-1}(z)|^k dz \right)^{\frac{1}{k}}, \tag{2}$$

where $F_\mu^{-1}$ and $F_\nu^{-1}$ are inverse cumulative distribution functions (CDFs) of $\mu$ and $\nu$, respectively. In practice, Wasserstein distances $W_k(\tilde{\mu}, \tilde{\nu})$ between one-dimensional empirical distributions $\tilde{\mu} = \frac{1}{N} \sum_{n=1}^{N} \delta_{x_n}$ and $\tilde{\nu} = \frac{1}{N} \sum_{n=1}^{N} \delta_{y_n}$ can be computed by sorting one-dimensional samples from empirical distributions, and evaluating the distances between sorted samples (Kolouri et al., 2019b):

$$W_k(\tilde{\mu}, \tilde{\nu}) = \left( \frac{1}{N} \sum_{n=1}^{N} |x_{I_x[n]} - y_{I_y[n]}|^k \right)^{\frac{1}{k}}, \tag{3}$$

where $N$ is the number of samples, $I_x[n]$ and $I_y[n]$ are the indices of sorted samples satisfying $x_{I_x[n]} \le x_{I_x[n+1]}$ and $y_{I_y[n]} \le y_{I_y[n+1]}$, respectively.

**Radon transform and generalized Radon transform:** The Radon transform (Radon, 1917) maps a function $f(\cdot) \in L^1(\mathbb{R}^d)$ to the space of functions defined over spaces of hyperplanes in $\mathbb{R}^d$. The Radon transform of $f(\cdot)$ is defined by line integrals of $f(\cdot)$ along all possible hyperplanes in $\mathbb{R}^d$:

$$\mathcal{R}f(t, \theta) = \int_{\mathbb{R}^d} f(x) \delta(t - \langle x, \theta \rangle) dx, \tag{4}$$

where $t \in \mathbb{R}$ and $\theta \in \mathbb{S}^{d-1}$ represent the parameters of hyperplanes $\{x \in \mathbb{R}^d \mid \langle x, \theta \rangle = t\}$, $\delta(\cdot)$ is the one-dimensional Dirac delta function, and $\langle \cdot, \cdot \rangle$ refers to the Euclidean inner product.

By replacing the inner product $\langle x, \theta \rangle$ in Equation (4) with $\beta(x, \theta)$, a specific family of functions named as *defining function* in (Kolouri et al., 2019a), the generalized Radon transform (GRT) (Beylkin, 1984) is defined as integrals of $f(\cdot)$ along hypersurfaces defined by $\{x \in \mathbb{R}^d \mid \beta(x, \theta) = t\}$:

$$\mathcal{G}f(t, \theta) = \int_{\mathbb{R}^d} f(x) \delta(t - \beta(x, \theta)) dx, \tag{5}$$

where $t \in \mathbb{R}$, $\theta \in \Omega_\theta$ and $\Omega_\theta$ is a compact set of all feasible $\theta$, e.g. $\Omega_\theta = \mathbb{S}^{d-1}$ for $\beta(x, \theta) = \langle x, \theta \rangle$. In particular, a function $\beta(x, \theta)$ defined on $\mathcal{X} \times (\mathbb{R}^d \setminus \{0\})$ with $\mathcal{X} \subseteq \mathbb{R}^d$ is called a defining function of GRTs if it satisfies conditions **H.1 – H.4** given in (Kolouri et al., 2019a).

For probability measures $\mu \in P_k(\mathbb{R}^d)$, the Radon transform and the GRT can be employed as push-forward operators, and the generated push-forward measures $\mathcal{R}_\mu = \mathcal{R} \# \mu$, $\mathcal{G}_\mu = \mathcal{G} \# \mu$ are defined as follows (Bonneel et al., 2015):

$$\mathcal{R}_\mu(t, \theta) = \int_{\mathbb{R}^d} \delta(t - \langle x, \theta \rangle) d\mu, \tag{6}$$

$$\mathcal{G}_\mu(t, \theta) = \int_{\mathbb{R}^d} \delta(t - \beta(x, \theta)) d\mu. \tag{7}$$

Notably, the Radon transform is a linear bijection (Helgason, 1980), and the sufficient conditions for GRTs to be bijective are provided in (Homan and Zhou, 2017).

**Sliced Wasserstein distance and generalized sliced Wasserstein distance:** By applying the Radon transform to $\mu$ and $\nu$ to obtain multiple projections, the sliced Wasserstein distance (SWD) decomposes the high-dimensional Wasserstein distance into multiple one-dimensional Wasserstein distances which can be efficiently evaluated (Bonneel et al., 2015). The $k$-SWD between $\mu$ and $\nu$ is defined by:

$$\mathrm{SWD}_k(\mu,\nu) = \left( \int_{\mathbb{S}^{d-1}} W_k^k \big( \mathcal{R}_\mu(\cdot,\theta), \mathcal{R}_\nu(\cdot,\theta) \big) d\theta \right)^{\frac{1}{k}}, \tag{8}$$

where the Radon transform $\mathcal{R}$ defined by Equation (6) is adopted as the measure push-forward operator. The GSWD generalizes the idea of SWD by slicing distributions with hypersurfaces rather than hyperplanes (Kolouri et al., 2019a). The GSWD is defined as:

$$\mathrm{GSWD}_k(\mu,\nu) = \left( \int_{\Omega_\theta} W_k^k \big( \mathcal{G}_\mu(\cdot,\theta), \mathcal{G}_\nu(\cdot,\theta) \big) d\theta \right)^{\frac{1}{k}}, \tag{9}$$

where the GRT $\mathcal{G}$ defined by Equation (7) is used as the measure push-forward operator. From Equation (3), with $L$ random projections and $N$ samples, the SWD and GSWD between $\mu$ and $\nu$ can be approximated by:

$$\mathrm{SWD}_k(\mu,\nu) \approx \left( \frac{1}{NL} \sum_{l=1}^{L} \sum_{n=1}^{N} |\langle x_{I_x^l[n]}, \theta_l \rangle - \langle y_{I_y^l[n]}, \theta_l \rangle|^k \right)^{\frac{1}{k}}, \tag{10}$$

$$\mathrm{GSWD}_k(\mu,\nu) \approx \left( \frac{1}{NL} \sum_{l=1}^{L} \sum_{n=1}^{N} |\beta(x_{I_x^l[n]}, \theta_l) - \beta(y_{I_y^l[n]}, \theta_l)|^k \right)^{\frac{1}{k}}, \tag{11}$$

where $I_x^l$ and $I_y^l$ are sequences consisting of the indices of sorted samples which satisfy $\langle x_{I_x^l[n]}, \theta_l \rangle \leq \langle x_{I_x^l[n+1]}, \theta_l \rangle$, $\langle y_{I_y^l[n]}, \theta_l \rangle \leq \langle y_{I_y^l[n+1]}, \theta_l \rangle$ in the SWD, and $\beta(x_{I_x^l[n]}, \theta_l) \leq \beta(x_{I_x^l[n+1]}, \theta_l)$, $\beta(y_{I_y^l[n]}, \theta_l) \leq \beta(y_{I_y^l[n+1]}, \theta_l)$ in the GSWD. The approximation error in estimating SWDs using Equation (10) is derived in (Nadjahi et al., 2020). It is proved in (Bonnotte, 2013) that the SWD is a valid distance metric. The GSWD is a valid metric except for its neural network variant (Kolouri et al., 2019a).

## 3 AUGMENTED SLICED WASSERSTEIN DISTANCES

In this section, we propose a new distance metric called the augmented sliced Wasserstein distance (ASWD), which embeds flexible nonlinear projections in its construction. We also provide an implementation recipe for the ASWD.

### 3.1 SPATIAL RADON TRANSFORM AND AUGMENTED SLICED WASSERSTEIN DISTANCE

In the definitions of the SWD and GSWD, the Radon transform (Radon, 1917) and the generalized Radon transform (GRT) (Beylkin, 1984) are used as the push-forward operator for projecting distributions to a one-dimensional space. However, it is not straightforward to design defining functions $\beta(x,\theta)$ for the GRT, since one needs to first check if $\beta(x,\theta)$ satisfies the conditions to be a defining function (Kolouri et al., 2019a; Beylkin, 1984), and then whether the corresponding GRT is bijective or not (Homan and Zhou, 2017). In practice, the assumption of the transform can be relaxed, as Theorem 1 shows that as long as the transform is injective, the corresponding ASWD metric is a valid distance metric.

To help us define the augmented sliced Wasserstein distance, we first introduce the *spatial Radon transform* which includes the Radon transform and the polynomial GRT as special cases (See Remark 5).

**Definition 1.** *Given a measurable injective mapping $g(\cdot): \mathbb{R}^d \to \mathbb{R}^{d_\theta}$ and a function $f(\cdot) \in L^1(\mathbb{R}^d)$, the spatial Radon transform of $f(\cdot)$ is defined as*

$$\mathcal{H}f(t,\theta;g) = \int_{\mathbb{R}^d} f(x)\delta(t - \langle g(x), \theta \rangle) dx, \tag{12}$$

*where $t \in \mathbb{R}$ and $\theta \in \mathbb{S}^{d_\theta-1}$ are the parameters of hypersurfaces $\{x \in \mathbb{R}^d \mid \langle g(x), \theta \rangle = t\}$.*

Similar to the Radon transform and the GRT, the spatial Radon transform can also be used to generate push-forward measure $\mathcal{H}_\mu = \mathcal{H}\#\mu$ for $\mu \in P_k(\mathbb{R}^d)$ as in Equations (6) and (7):

$$\mathcal{H}_\mu(t,\theta;g) = \int_{\mathbb{R}^d} \delta(t - \langle g(x),\theta \rangle)d\mu. \tag{13}$$

**Remark 1.** *Note that the spatial Radon transform can be interpreted as applying the vanilla Radon transform to $\hat{\mu}_g$, where $\hat{\mu}_g$ refers to the push-forward measure $g_\#\mu$, i.e given a measurable injective mapping $g(\cdot): \mathbb{R}^d \to \mathbb{R}^{d_\theta}$, the spatial Radon transform defined by Equation (13) can be rewritten as:*

$$\begin{aligned}
\mathcal{H}_\mu(t,\theta;g) &= E_{x\sim\mu}[\delta(t - \langle g(x),\theta \rangle)], \\
&= E_{\hat{x}\sim\hat{\mu}_g}[\delta(t - \langle \hat{x},\theta \rangle)] \\
&= \int \delta(t - \langle \hat{x},\theta \rangle)d\hat{\mu}_g \\
&= \mathcal{R}_{\hat{\mu}_g}(t,\theta).
\end{aligned} \tag{14}$$

*Hence the spatial Radon transform inherits the theoretical properties of the Radon transform and incorporates nonlinear projections through $g(\cdot)$.*

In what follows, we use $\mu \equiv \nu$ to denote probability measures $\mu,\nu \in P_k(\mathbb{R}^d)$ that satisfy $\mu(\mathcal{X}) = \nu(\mathcal{X})$ for $\forall \mathcal{X} \subseteq \mathbb{R}^d$.

**Lemma 1.** *Given an injective mapping $g(\cdot): \mathbb{R}^d \to \mathbb{R}^{d_\theta}$ and two probability measures $\mu,\nu \in P(\mathbb{R}^d)$, for all $t \in \mathbb{R}$ and $\theta \in \mathbb{S}^{d_\theta-1}$, $\mathcal{H}_\mu(t,\theta;g) \equiv \mathcal{H}_\nu(t,\theta;g)$ if and only if $\mu \equiv \nu$, i.e. the spatial Radon transform is an injection on $P_k(\mathbb{R}^d)$. Moreover, the spatial Radon transform is an injection on $P_k(\mathbb{R}^d)$ if and only if the mapping $g(\cdot)$ is an injection.*

See Appendix A for the proof of Lemma 1.

We now introduce the augmented sliced Wasserstein distance, by utilizing the spatial Radon transform as the measure push-forward operator:

**Definition 2.** *Given two probability measures $\mu,\nu \in P_k(\mathbb{R}^d)$ and an injective mapping $g(\cdot): \mathbb{R}^d \to \mathbb{R}^{d_\theta}$, the augmented sliced Wasserstein distance (ASWD) of order $k \in [1,+\infty)$ is defined as:*

$$\mathrm{ASWD}_k(\mu,\nu;g) = \left( \int_{\mathbb{S}^{d_\theta-1}} W_k^k\big(\mathcal{H}_\mu(\cdot,\theta;g),\mathcal{H}_\nu(\cdot,\theta;g)\big)d\theta \right)^{\frac{1}{k}}, \tag{15}$$

*where $\theta \in \mathbb{S}^{d_\theta-1}$, $W_k$ is the $k$-Wasserstein distance defined by Equation (1), and $\mathcal{H}$ refers to the spatial Radon transform defined by Equation (13).*

**Remark 2.** *Following the connection between the spatial Radon transform and the vanilla Radon transform as shown in Equation (14), the ASWD can be rewritten as:*

$$\begin{aligned}
\mathrm{ASWD}_k(\mu,\nu;g) &= \left( \int_{\mathbb{S}^{d_\theta-1}} W_k^k\big(\mathcal{R}_{\hat{\mu}_g}(\cdot,\theta),\mathcal{R}_{\hat{\nu}_g}(\cdot,\theta)\big)d\theta \right)^{\frac{1}{k}} \\
&= \mathrm{SWD}_k(\hat{\mu}_g,\hat{\nu}_g),
\end{aligned} \tag{16}$$

*where $\hat{\mu}_g$ and $\hat{\nu}_g$ are probability measures on $\mathbb{R}^{d_\theta}$ which satisfy $g(x) \sim \hat{\mu}_g$ for $x \sim \mu$ and $g(y) \sim \hat{\nu}_g$ for $y \sim \nu$.*

**Theorem 1.** *The augmented sliced Wasserstein distance (ASWD) of order $k \in [1,+\infty)$ defined by Equation (15) with a mapping $g(\cdot): \mathbb{R}^d \to \mathbb{R}^{d_\theta}$ is a metric on $P_k(\mathbb{R}^d)$ if and only if $g(\cdot)$ is injective.*

The proof of Theorem 1 is provided in Appendix C. Theorem 1 shows that the ASWD is a metric given a fixed injective mapping $g(\cdot)$. In practical applications, the mapping $g(\cdot)$ needs to be optimized to project samples onto discriminating hypersurfaces. We show in Corollary 1.1 that the ASWD between $\mu$ and $\nu$ with the optimized $g(\cdot)$ is also a metric under mild conditions.

**Corollary 1.1.** *The augmented sliced Wasserstein distance (ASWD) of order $k \in [1,+\infty)$ between two probability measures $\mu,\nu \in P_k(\mathbb{R}^d)$ defined by Equation (15) with the optimal mapping*

$$g^*(\cdot) = \underset{g}{\mathrm{argmax}}\{\mathrm{ASWD}_k(\mu,\nu;g) - L(\mu,\nu,\lambda;g)\} \tag{17}$$

*is a metric on $P_k(\mathbb{R}^d)$, where $L(\mu,\nu,\lambda;g) = \lambda(\mathbb{E}_{x\sim\mu}^{\frac{1}{k}}\big[||g(x)||_2^k\big] + \mathbb{E}_{y\sim\nu}^{\frac{1}{k}}\big[||g(y)||_2^k\big])$ for $\lambda \in (1,+\infty)$.*

The proof of Corollary 1.1 is provided in Appendix D.

**Remark 3.** *Corollary 1.1 shows that given measures $\mu_1, \mu_2, \mu_3 \in P_k(\mathbb{R}^d)$, the triangle inequality holds for the ASWD when $g(\cdot)$ is optimized for each pair of measures, as shown in Appendix D. It is worth noting that $\lambda > 1$ is a sufficient condition for the ASWD to be a metric – as further discussed in Remark 6, $0 < \lambda \leq 1$ can also lead to finite $||g(x)||_2$ in various scenarios, resulting in valid metrics. The discussion on the impact of $\lambda$ on the performance of the ASWD in practice can be found in Appendix G.2.*

### 3.2 NUMERICAL IMPLEMENTATION

We discuss in this section how to realize injective mapping $g(\cdot)$ with *neural networks* due to their expressiveness and optimize it with gradient based methods.

**Injective neural networks:** As stated in Lemma 1 and Theorem 1, the injectivity of $g(\cdot)$ is the *sufficient and necessary* condition for the ASWD being a valid metric. Thus we need specific architecture designs on implementing $g(\cdot)$ by neural networks. One option is the family of invertible neural networks (Behrmann et al., 2019; Karami et al., 2019), which are both injective and surjective. However, the running cost of those models is usually much higher than that of vanilla neural networks. We propose an alternative approach by concatenating the input $x$ of an arbitrary neural network to its output $\phi_\omega(x)$:

$$g_\omega(x) = [x, \phi_\omega(x)]. \tag{18}$$

It is trivial to show that $g_\omega(x)$ is injective, since different inputs will lead to different outputs. Although embarrassingly simple, this idea of concatenating the input and output of neural networks has found success in preserving information with dense blocks in the DenseNet (Huang et al., 2017), where the input of each layer is injective to the output of all preceding layers.

**Optimization objective:** We aim to slice distributions with maximally discriminating hypersurfaces between two distributions while avoiding the projected samples being arbitrarily large, so that the projected samples between the compared distributions are finite and most dissimilar regarding the ASWD, as shown in Figure 1. Similar ideas have been employed to identify important projection directions (Deshpande et al., 2019; Kolouri et al., 2019a; Paty and Cuturi, 2019) or a discriminative ground metric (Salimans et al., 2018) in optimal transport metrics. For the ASWD, the parameterized injective neural network $g_\omega(\cdot)$ is optimized by maximizing the following objective:

$$\mathcal{L}(\mu, \nu; g_\omega, \lambda) = \left( \int_{\mathbb{S}^{d_\theta - 1}} W_k^k \big( \mathcal{H}_\mu(\cdot, \theta; g_\omega), \mathcal{H}_\nu(\cdot, \theta; g_\omega) \big) d\theta \right)^{\frac{1}{k}} - L(\mu, \nu, \lambda; g_\omega), \tag{19}$$

where $\lambda > 0$ and the regularization term $L(\mu, \nu, \lambda; g_\omega) = \lambda (\mathbb{E}_{x \sim \mu}^{\frac{1}{k}} \big[ ||g_\omega(x)||_2^k \big] + \mathbb{E}_{y \sim \nu}^{\frac{1}{k}} \big[ ||g_\omega(y)||_2^k \big])$ on the magnitude of the neural network's output is used, otherwise the projections may be arbitrarily large.

**Remark 4.** *The regularization coefficient $\lambda$ adjusts the introduced non-linearity in the evaluation of the ASWD by controlling the norm of $\phi_\omega(\cdot)$ in Equation (18). In particular, when $\lambda \to \infty$, the nonlinear term $\phi_\omega(\cdot)$ shrinks to $0$. The intrinsic dimension of the augmented space, i.e. the number of non-zero dimensions in the augmented space, is hence explicitly controlled by the flexible choice of $\phi_\omega(\cdot)$ and implicitly regularized by $L(\mu, \nu, \lambda; g_\omega)$.*

By plugging the optimized $g_{\omega, \lambda}^*(\cdot) = \underset{g_\omega}{\operatorname{argmax}}(\mathcal{L}(\mu, \nu; g_\omega, \lambda))$ into Equation (15), we obtain the empirical version of the ASWD. Pseudocode is provided in Appendix E.

## 4 RELATED WORK

Recent work on slice-based Wasserstein distances mainly focused on improving their projection efficiency, leading to a reduced number of projections needed to capture the structure of data distributions (Kolouri et al., 2019a; Nguyen et al., 2021). The GSWD proposes using nonlinear projections to achieve this goal, and it has been proved to be a valid distance metric if and only if they adopt injective GRTs, which only include the circular functions and a finite number of harmonic polynomial functions with odd degrees as their feasible defining functions (Ehrenpreis, 2003). While the GSWD has shown impressive performance in various applications (Kolouri et al., 2019a), its defining function is restricted to the aforementioned limited class of candidates. In addition, the

selection of defining function is usually task-dependent and needs domain knowledge, and the impact on performance from different defining functions is still unclear.

To tackle those limitations, (Kolouri et al., 2019a) proposed the GSWD-NN, which *directly* takes the outputs of a neural network as its projection results without using the standard Radon transform or GRTs. However, this brings three side effects: 1) The number of projections, which equals the number of nodes in the neural network's output layer, is fixed, thus new neural networks are needed if one wants to change the number of projections. 2) There is no random projections involved in the GSWD-NN, as the projection results are determined by the inputs and weights of the neural network. 3) The GSWD-NN is a pseudo-metric since it uses a vanilla neural network, rather than Radon transform or GRTs, as its push-forward operator. Therefore, the GSWD-NN does not fit into the theoretical framework of GSWD and does not inherit its geometric properties.

Another notable variant of the SWD is the distributional sliced Wasserstein distance (DSWD) (Nguyen et al., 2021). By finding a distribution of projections that maximizes the expected distances over these projections, the DSWD can slice distributions from multiple directions while having high projection efficiency. Injective GRTs are also used to extend the DSWD to the distributional generalized sliced Wasserstein distance (DGSWD) (Nguyen et al., 2021). Experiment results show that the DSWD and the DGSWD have superior performance in generative modelling tasks (Nguyen et al., 2021). However, neither the DSWD nor the DGSWD have solved the problem with the GSWD, i.e. they are still not able to produce nonlinear projections adaptively.

Our contribution differs from previous work in three ways: 1) The ASWD is data-adaptive, i.e. the hyper-surfaces where high-dimensional distributions are projected onto can be learned from data. This implies one does not need to specify a defining function from limited choices. 2) Unlike GSWD-NN, the ASWD takes a novel direction to incorporate neural networks into the framework of sliced-based Wasserstein distances while maintaining the properties of sliced Wasserstein distances. 3) Previous work on introducing nonlinear projections into Radon transform either is restricted to only a few candidates of defining functions (GRTs) or breaks the framework of Radon transforms (neural networks in GSWD-NN), in contrast, the spatial Radon transform provides a novel way of defining nonlinear Radon-type transforms.

## 5 EXPERIMENTS

In this section, we describe the experiments that we have conducted to evaluate performance of the proposed distance metric. The GSWD leads to the best performance in a sliced Wasserstein flow problem reported in (Kolouri et al., 2019a) and the DSWD outperforms the compared methods in the generative modeling task examined in (Nguyen et al., 2021) on CIFAR 10 (Krizhevsky, 2009), CelebA (Liu et al., 2015), and MNIST (LeCun et al., 1998) datasets (Appendix H.2). Hence, we compare performance of the ASWD with the state-of-the-art distance metrics in the same examples and report results as below[1]. We provide additional experiment results in the appendices, including a sliced Wasserstein autoencoder (SWAE) (Kolouri et al., 2019b) using the ASWD (Appendix I), image color transferring (Appendix J) and sliced Wasserstein barycenters (Appendix K).

To examine the robustness of the ASWD, throughout the experiments, we adopt the injective network architecture given in Equation (18) and set $\phi_\omega$ to be a single fully-connected layer neural network whose output dimension equals its input dimension, with a ReLU layer as its activation function. The order $k$ is set to be 2 in all experiments.

### 5.1 SLICED WASSERSTEIN FLOWS

We first consider the problem of evolving a source distribution $\mu$ to a target distribution $\nu$ by minimizing slice-based Wasserstein distances between $\mu$ and $\nu$ in the sliced Wasserstein flow task reported in (Kolouri et al., 2019a).

$$\partial_t \mu_t = -\nabla \text{SWD}(\mu_t, \nu), \tag{20}$$

where $\mu_t$ refers to the updated source distribution at each iteration $t$. The SWD in Equation (20) can be replaced by other sliced-Wasserstein distances to be evaluated. As in (Kolouri et al., 2019a), the 2-Wasserstein distance was used as the metric for evaluating performance of different distance metrics in this task. The set of hyperparameter values used in this experiment can be found in Appendix F.1.

---

[1]Code to reproduce experiment results is available at: https://github.com/xiongjiechen/ASWD.

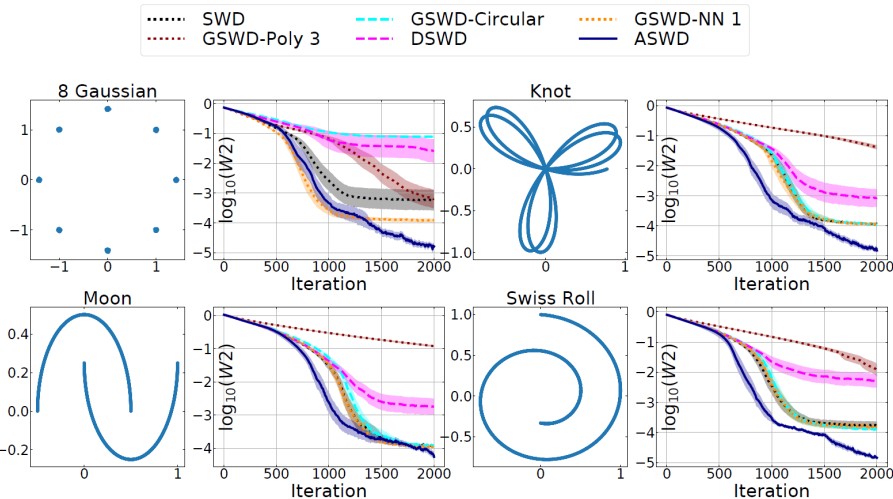

Figure 2: The first and third columns are target distributions. The second and fourth columns are log 2-Wasserstein distances between the target distribution and the source distribution. The horizontal axis show the number of training iterations. Solid lines and shaded areas represent the average values and 95% confidence intervals of log 2-Wasserstein distances over 50 runs. A more extensive set of experimental results can be found in Appendix G.1.

Without loss of generality, we initialize $\mu_0$ to be the standard normal distribution $\mathcal{N}(0,I)$. We repeat each experiment 50 times and record the 2-Wasserstein distance between $\mu$ and $\nu$ at every iteration. In Figure 2, we plot the 2-Wasserstein distances between the source and target distributions as a function of the training epochs and the 8-Gaussian, the Knot, the Moon, and the Swiss roll distributions are respective target distributions. For clarity, Figure 2 displays the experiment results from the 6 best performing distance metrics, including the ASWD, the DSWD, the SWD, the GSWD-NN 1, which directly generates projections through a one layer MLP, as well as the GSWD with the polynomial of degree 3, circular defining functions, out of the 12 distance metrics we compared.

We observe from Figure 2 that the ASWD not only leads to smaller 2-Wasserstein distances, but also converges faster by achieving better results with fewer iterations than the other methods in these four target distributions. A complete set of experimental results with 12 compared distance metrics and 8 target distributions are included in Appendix G.1. The ASWD outperforms the compared state-of-the-art sliced-based Wasserstein distance metrics with 7 out of the 8 target distributions except for the 25-Gaussian. This is achieved through the simple injective network architecture given in Equation (18) and a one layer fully-connected neural network with equal input and output dimensions throughout the experiments. In addition, ablation study is conducted to study the effect of injective neural networks, the regularization coefficient $\lambda$, the choice of the dimensionality $d_\theta$ of the augmented space, and the optimization of hypersurfaces in the ASWD. Details can be found in Appendix G.2.

## 5.2 GENERATIVE MODELING

In this experiment, we use sliced-based Wasserstein distances for a generative modeling task described in (Nguyen et al., 2021). The task is to generate images using generative adversarial networks (GANs) (Goodfellow et al., 2014) trained on either the CIFAR10 dataset (64×64 resolution) (Krizhevsky, 2009) or the CelebA dataset (64×64 resolution) (Liu et al., 2015). Denote the hidden layer and the output layer of the discriminator by $h_\psi$ and $D_\Psi$, and the generator by $G_\Phi$, we train GAN models with the following objectives:

$$\min_{\Phi} \text{SWD}(h_\psi(p_r), h_\psi(G_\Phi(p_z))), \tag{21}$$

$$\max_{\Psi,\psi} \mathbb{E}_{x \sim p_r}[\log(D_\Psi(h_\psi(x)))] + \mathbb{E}_{z \sim p_z}[\log(1 - D_\Psi(h_\psi(G_\Phi(z))))], \tag{22}$$

where $p_z$ is the prior of latent variable $z$ and $p_r$ is the distribution of real data. The SWD in Equation (21) is replaced by the ASWD and other variants of the SWD to compare their performance. The

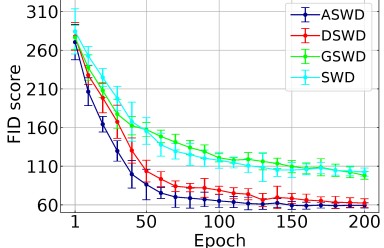 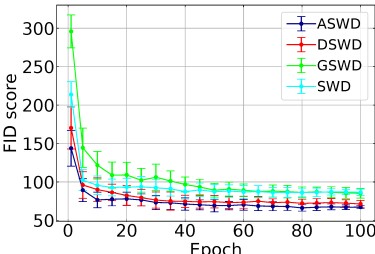

Figure 3: FID scores of generative models trained with different metrics on CIFAR10 (left) and CelebA (right) datasets with $L = 1000$ projections. The error bar represents the standard deviation of the FID scores at the specified training epoch among 10 simulation runs.

Table 1: FID scores of generative models trained with different distance metrics. Smaller scores indicate better image qualities. $L$ is the number of projections, we run each experiment 10 times and report the average values and standard errors of FID scores for CIFAR10 dataset and CELEBA dataset. The running time per training iteration for one batch containing 512 samples is computed based on a computer with an Intel (R) Xeon (R) Gold 5218 CPU 2.3 GHz and 16GB of RAM, and a RTX 6000 graphic card with 22GB memories.

| | CIFAR10 | | | | | | | |
|---|---|---|---|---|---|---|---|---|
| | SWD (Bonneel et al., 2015) | | GSWD (Kolouri et al., 2019a) | | DSWD (Nguyen et al., 2021) | | ASWD | |
| $L$ | FID | $t$ (s/it) | FID | $t$ (s/it) | FID | $t$ (s/it) | FID | $t$ (s/it) |
| 10 | 121.4±7.0 | 0.34 | 108.3±5.6 | 0.41 | 74.2 ± 3.1 | 0.55 | **65.7±3.2** | 0.58 |
| 100 | 104.6±5.2 | 0.35 | 105.2±3.2 | 0.74 | 66.5 ± 3.9 | 0.57 | **62.5±1.9** | 0.60 |
| 1000 | 102.3±5.3 | 0.36 | 98.2±5.1 | 2.22 | 62.3 ± 5.7 | 1.30 | **59.3±3.2** | 1.38 |
| | CELEBA | | | | | | | |
| 10 | 94.8±2.5 | 0.35 | 95.1±4.2 | 0.40 | 86.0 ± 1.4 | 0.53 | **81.2±1.3** | 0.59 |
| 100 | 88.7±5.7 | 0.36 | 86.7±3.5 | 0.75 | 76.1 ± 3.5 | 0.55 | **73.2±2.6** | 0.61 |
| 1000 | 86.5±4.1 | 0.38 | 85.2±6.3 | 2.19 | 71.3±4.7 | 1.28 | **67.4±2.1** | 1.38 |

GSWD with the polynomial defining function and the DGSWD is not included in this experiment due to its excessively high computational cost in high-dimensional space. The *Fréchet Inception Distance* (FID score) (Heusel et al., 2017) is used to assess the quality of generated images. More details on the network structures and the parameter setup used in this experiment are available in Appendix F.2.

We run 200 and 100 training epochs to train the GAN models on the CIFAR10 and the CelebA dataset, respectively. Each experiment is repeated for 10 times and results are reported in Table 1. With the same number of projections and a similar computation cost, the ASWD leads to significantly improved FID scores among all evaluated distances metrics on both datasets, which implies that images generated with the ASWD are of higher qualities. Figure 3 plots the FID scores recorded during the training process. The GAN model trained with the ASWD exhibits a faster convergence as it reaches smaller FID scores with fewer epochs. Randomly selected samples of generated images are presented in Appendix H.1.

## 6 CONCLUSION

We proposed a novel variant of the sliced Wasserstein distance, namely the augmented sliced Wasserstein distance (ASWD), which is flexible, has a high projection efficiency, and generalizes well. The ASWD adaptively updates the hypersurfaces used to slice compared distributions by learning from data. We proved that the ASWD is a valid distance metric and presented its numerical implementation. We reported empirical performance of the ASWD over state-of-the-art sliced Wasserstein metrics in various numerical experiments. We showed that ASWD with a simple injective neural network architecture can lead to the smallest distance errors over the majority of datasets in a sliced Wasserstein flow task and superior performance in generative modeling tasks involving GANs and VAEs. We have also evaluated the applications of the ASWD in downstream tasks including color transferring and Wasserstein barycenters. What remains to be explored includes the topological properties of the ASWD. We leave this topic as an interesting future research direction.

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

## APPENDIX A    PROOF OF THE LEMMA 1

We prove that the spatial Radon transform defined with a measurable mapping $g(\cdot):\mathbb{R}^d\to\mathbb{R}^{d_\theta}$ is an injection on $P_k(\mathbb{R}^d)$ if and only if $g(\cdot)$ is injective. In the following contents, we use $P_k(\mathbb{R}^d)$ to denote a set of Borel probability measures with finite $k$-th moment on $\mathbb{R}^d$, and $f_1\equiv f_2$ is used to denote functions $f_1(\cdot):X\to\mathbb{R}$ and $f_2(\cdot):X\to\mathbb{R}$ that satisfy $f_1(x)=f_2(x)$ for $\forall x\in X$, and $f_1\not\equiv f_2$ is used to denote functions $f_1(\cdot):X\to\mathbb{R}$ and $f_2(\cdot):X\to\mathbb{R}$ that satisfy $f_1(x)\neq f_2(x)$ for certain $x\in X$. In addition, for probability measures $\mu$ and $\nu$, we use $\mu\equiv\nu$ to denote $\mu,\nu\in P_k(\mathbb{R}^d)$ that satisfy $\mu(\mathcal{X})=\nu(\mathcal{X})$ for $\forall\mathcal{X}\subseteq\mathbb{R}^d$, and $\mu\not\equiv\nu$ to denote $\mu,\nu\in P_k(\mathbb{R}^d)$ that satisfy $\mu(\mathcal{X})\neq\nu(\mathcal{X})$ for certain $\mathcal{X}\subseteq\mathbb{R}^d$.

*Proof.* By using proof by contradiction, we first prove that if $g(\cdot)$ is injective, the corresponding spatial Radon transform is injective. If the spatial Radon transform defined with an injective mapping $g(\cdot):\mathbb{R}^d\to\mathbb{R}^{d_\theta}$ is not injective, there exist $\mu,\nu\in P_k(\mathbb{R}^d)$, $\mu\not\equiv\nu$, such that $\mathcal{H}_\mu(t,\theta;g)\equiv\mathcal{H}_\nu(t,\theta;g)$ for $\forall t\in\mathbb{R}$ and $\forall\theta\in\mathbb{S}^{d_\theta-1}$.

From Equation (14), for $\forall t\in\mathbb{R}$ and $\forall\theta\in\mathbb{S}^{d_\theta-1}$, the spatial Radon transform of $\mu$ can be written as:

$$\mathcal{H}_\mu(t,\theta;g)=\mathcal{R}_{\hat{\mu}_g}(t,\theta),\tag{23}$$

$$\mathcal{H}_\nu(t,\theta;g)=\mathcal{R}_{\hat{\nu}_g}(t,\theta),\tag{24}$$

where $\hat{\mu}_g$ and $\hat{\nu}_g$ refer to the push-forward measures $g_{\#}\mu$ and $g_{\#}\nu$, respectively. From the assumption $\mathcal{H}_\mu(t,\theta;g)\equiv\mathcal{H}_\nu(t,\theta;g)$ and Equations (23) and (24), we know $\mathcal{R}_{\hat{\mu}_g}(t,\theta)\equiv\mathcal{R}_{\hat{\nu}_g}(t,\theta)$ for $\forall t\in\mathbb{R}$ and $\forall\theta\in\mathbb{S}^{d_\theta-1}$, which implies $\hat{\mu}_g\equiv\hat{\nu}_g$ since the Radon transform is injective.

Since $g(\cdot)$ is injective, for all measurable $\mathcal{X}\subseteq\mathbb{R}^d$, $x\in\mathcal{X}$ if and only if $\hat{x}=g(x)\in g(\mathcal{X})$, which implies $P(x\in\mathcal{X})=P(\hat{x}\in g(\mathcal{X}))$, $P(y\in\mathcal{X})=P(\hat{y}\in g(\mathcal{X}))$. Therefore,

$$\int_{g(\mathcal{X})}d\hat{\mu}_g=\int_{\mathcal{X}}d\mu,\tag{25}$$

$$\int_{g(\mathcal{X})}d\hat{\nu}_g=\int_{\mathcal{X}}d\nu.\tag{26}$$

Since $\hat{\mu}_g\equiv\hat{\nu}_g$, from Equations (25) and (26) we have: $\int_{\mathcal{X}}d\mu=\int_{\mathcal{X}}d\nu$ for $\forall\mathcal{X}\subseteq\mathbb{R}^d$. Hence, for $\forall\mathcal{X}\subseteq\mathbb{R}^d$:

$$\int_{\mathcal{X}}d(\mu-\nu)=0,\tag{27}$$

which implies $\mu\equiv\nu$, contradicting with the assumption $\mu\not\equiv\nu$. Therefore, if $\mathcal{H}_\mu\equiv\mathcal{H}_\nu$, $\mu\equiv\nu$. In addition, from the definition of the spatial Radon transform in Equation (13), it is trivial to show that if $\mu\equiv\nu$, $\mathcal{H}_\mu(t,\theta;g)\equiv\mathcal{H}_\nu(t,\theta;g)$. Therefore, $\mathcal{H}_\mu\equiv\mathcal{H}_\nu$ if and only if $\mu\equiv\nu$, i.e. the spatial Radon transform $\mathcal{H}$ defined with an injective mapping $g(\cdot):\mathbb{R}^d\to\mathbb{R}^{d_\theta}$ is injective.

We now prove that if the spatial Radon transform defined with a mapping $g(\cdot):\mathbb{R}^d\to\mathbb{R}^{d_\theta}$ is injective, $g(\cdot)$ must be injective. Again, we use proof by contradiction. If $g(\cdot)$ is not injective, there exist $x_0,y_0\in\mathbb{R}^d$ such that $x_0\neq y_0$ and $g(x_0)=g(y_0)$. For Dirac measures $\mu_1$ and $\nu_1$ defined by $\mu_1(\mathcal{X})=\int_{\mathcal{X}}\delta(x-x_0)dx$ and $\nu_1(\mathcal{Y})=\int_{\mathcal{Y}}\delta(y-y_0)dy$, where $\mathcal{X},\mathcal{Y}\subseteq\mathbb{R}^d$, we know $\mu_1\not\equiv\nu_1$ as $x_0\neq y_0$.

We define variables $x\sim\mu_1$ and $y\sim\nu_1$. Then for variables $\hat{x}=g(x)\sim\mu_2$ and $\hat{y}=g(y)\sim\nu_2$, where $\mu_2$ and $\nu_2$ are push-forward measures $g_{\#}\mu_1$ and $g_{\#}\nu_1$, it is trivial to derive for $\forall\mathcal{X},\mathcal{Y}\subseteq\mathbb{R}^d$

$$\mu_2(g(\mathcal{X}))=\int_{g(\mathcal{X})}\delta(\hat{x}-g(x_0))d\hat{x},\tag{28}$$

$$\nu_2(g(\mathcal{Y}))=\int_{g(\mathcal{Y})}\delta(\hat{y}-g(y_0))d\hat{y},\tag{29}$$

which implies $\mu_2\equiv\nu_2$ as $g(x_0)=g(y_0)$.

From Equations (23), (24), (28) and (29), for $\forall t\in\mathbb{R}$ and $\forall\theta\in\mathbb{S}^{d_\theta-1}$:

$$\begin{aligned}\mathcal{H}_{\mu_1}(t,\theta;g)&=\mathcal{R}_{\mu_2}(t,\theta),\\&=\mathcal{R}_{\nu_2}(t,\theta),\\&=\mathcal{H}_{\nu_1}(t,\theta;g),\end{aligned}\tag{30}$$

contradicting with the assumption that the spatial Radon transform is injective. Therefore, if the spatial Radon transform is injective, $g(\cdot)$ must be injective. We conclude that the spatial Radon transform is injective if and only if the mapping $g(\cdot)$ is an injection on $P_k(\mathbb{R}^d)$. □

## APPENDIX B   SPECIAL CASES OF SPATIAL RADON TRANSFORMS

**Remark 5.** *The spatial Radon transform degenerates to the vanilla Radon transform when the mapping $g(\cdot)$ is an identity mapping. In addition, the spatial Radon transform is equivalent to the polynomial GRT (Ehrenpreis, 2003) when $g(x) = (x^{\alpha_1}, ..., x^{\alpha_{d_\alpha}})$, where $\alpha$ is multi-indices $\alpha_i = (\eta_{i,1}, ..., \eta_{i,d}) \in \mathbb{N}^d$ satisfying $|\alpha_i| = \sum_{j=1}^d \eta_{i,j} = m$. $m$ is the degree of the polynomial function, $x^{\alpha_i} = \prod_{j=1}^d x_j^{\eta_{i,j}}$ given an input $x = (x_1, ..., x_d) \in \mathbb{R}^d$, and $d_\alpha$ is the number of all possible multi-indices $\alpha_i$ that satisfies $|\alpha_i| = m$.*

We provide a proof for the claim in Remark 5.

*Proof.* Given a probability measure $\mu \in P(\mathbb{R}^d)$, the spatial Radon transform of $\mu$ is defined as:

$$\mathcal{H}_\mu(t, \theta; g) = \int_{\mathbb{R}^d} \delta(t - \langle g(x), \theta \rangle) d\mu, \tag{31}$$

where $t \in \mathbb{R}$ and $\theta \in \mathbb{S}^{d_\theta - 1}$ are the parameters of hypersurfaces in $\mathbb{R}^d$. When the mapping $g(\cdot)$ is an identity mapping, i.e. $g(x) = x$ for $\forall x \in \mathbb{R}^d$, the spatial Radon transform degenerates to the vanilla Radon transform:

$$\mathcal{H}_\mu(t, \theta; g) = \int_{\mathbb{R}^d} \delta(t - \langle x, \theta \rangle) d\mu$$
$$= \mathcal{R}_\mu(t, \theta). \tag{32}$$

For GRTs, (Ehrenpreis, 2003) provides a class of injective GRTs named polynomial GRTs by adopting homogeneous polynomial functions with an odd degree $m$ as the defining function:

$$\mathcal{G}_\mu(t, \theta) = \int_{\mathbb{R}^d} \delta(t - \sum_{i=1}^{d_\alpha} \theta_i x^{\alpha_i}) d\mu,$$
$$\text{s.t.} |\alpha_i| = m, \tag{33}$$

where $\alpha_i = (\eta_{i,1}, ..., \eta_{i,d}) \in \mathbb{N}^d$, $|\alpha_i| = \sum_{j=1}^d \eta_{i,j}$, $x^{\alpha_i} = \prod_{j=1}^d x_j^{\eta_{i,j}}$ for $x = (x_1, ..., x_d) \in \mathbb{R}^d$, $d_\alpha$ is the number of all possible multi-indices $\alpha_i$ that satisfies $|\alpha_i| = m$, and $\theta = (\theta_1, ..., \theta_{d_\alpha}) \in \mathbb{S}^{d_\alpha - 1}$.

In spatial Radon transform, for $\forall x \in \mathbb{R}^d$, when the mapping $g(\cdot)$ is defined as:

$$g(x) = (x^{\alpha_1}, ..., x^{\alpha_{d_\alpha}}), \tag{34}$$

the spatial Radon transform is equivalent to the polynomial GRT defined in Equation (33):

$$\mathcal{H}_\mu(t, \theta; g) = \int_{\mathbb{R}^d} \delta(t - \langle g(x), \theta \rangle) d\mu$$
$$= \int_{\mathbb{R}^d} \delta(t - \sum_{i=1}^{d_\alpha} \theta_i x^{\alpha_i}) d\mu. \tag{35}$$

$\square$

## APPENDIX C    PROOF OF THEOREM 1

We provide a proof that the ASWD defined with a mapping $g(\cdot):\mathbb{R}^d \to \mathbb{R}^{d_\theta}$ is a metric on $P_k(\mathbb{R}^d)$, if and only if $g(\cdot)$ is injective. In what follows, we denote a set of Borel probability measures with finite $k$-th moment on $\mathbb{R}^d$ by $P_k(\mathbb{R}^d)$, and use $\mu, \nu \in P_k(\mathbb{R}^d)$ to refer to two probability measures defined on $\mathbb{R}^d$.

*Proof.* **Symmetry:** Since the $k$-Wasserstein distance is a metric thus symmetric (Villani, 2008):

$$W_k\big(\mathcal{H}_\mu(\cdot,\theta;g),\mathcal{H}_\nu(\cdot,\theta;g)\big) = W_k\big(\mathcal{H}_\nu(\cdot,\theta;g),\mathcal{H}_\mu(\cdot,\theta;g)\big). \tag{36}$$

Therefore,

$$\begin{aligned}
\text{ASWD}_k(\mu,\nu;g) &= \left(\int_{\mathbb{S}^{d_\theta-1}} W_k^k\big(\mathcal{H}_\mu(\cdot,\theta;g),\mathcal{H}_\nu(\cdot,\theta;g)\big)d\theta\right)^{\frac{1}{k}} \\
&= \left(\int_{\mathbb{S}^{d_\theta-1}} W_k^k\big(\mathcal{H}_\nu(\cdot,\theta;g),\mathcal{H}_\mu(\cdot,\theta;g)\big)d\theta\right)^{\frac{1}{k}} = \text{ASWD}_k(\nu,\mu;g).
\end{aligned}$$

**Triangle inequality:** Given an injective mapping $g(\cdot) : \mathbb{R}^d \to \mathbb{R}^{d_\theta}$ and probability measures $\mu_1, \mu_2, \mu_3 \in P_k(\mathbb{R}^d)$, since the $k$-Wasserstein distance satisfies the triangle inequality (Villani, 2008), the following inequality holds:

$$\begin{aligned}
\text{ASWD}_k(\mu_1,\mu_3;g) &= \left(\int_{\mathbb{S}^{d_\theta-1}} W_k^k\big(\mathcal{H}_{\mu_1}(\cdot,\theta;g),\mathcal{H}_{\mu_3}(\cdot,\theta;g)\big)d\theta\right)^{\frac{1}{k}} \\
&\leq \left(\int_{\mathbb{S}^{d_\theta-1}} \big(W_k(\mathcal{H}_{\mu_1}(\cdot,\theta;g),\mathcal{H}_{\mu_2}(\cdot,\theta;g)) \right. \\
&\qquad \left. + W_k(\mathcal{H}_{\mu_2}(\cdot,\theta;g),\mathcal{H}_{\mu_3}(\cdot,\theta;g))\big)^k d\theta\right)^{\frac{1}{k}} \\
&\leq \left(\int_{\mathbb{S}^{d_\theta-1}} W_k^k\big(\mathcal{H}_{\mu_1}(\cdot,\theta;g),\mathcal{H}_{\mu_2}(\cdot,\theta;g)\big)d\theta\right)^{\frac{1}{k}} \\
&\qquad + \left(\int_{\mathbb{S}^{d_\theta-1}} W_k^k\big(\mathcal{H}_{\mu_2}(\cdot,\theta;g),\mathcal{H}_{\mu_3}(\cdot,\theta;g)\big)d\theta\right)^{\frac{1}{k}} \\
&= \text{ASWD}_k(\mu_1,\mu_2;g) + \text{ASWD}_k(\mu_2,\mu_3;g),
\end{aligned}$$

where the second inequality is due to the Minkowski inequality in $L^k(\mathbb{S}^{d_\theta-1})$.

**Identity of indiscernibles:** Since $W_k(\mu,\mu)=0$ for $\forall \mu \in P_k(\mathbb{R}^d)$, we have

$$\text{ASWD}_k(\mu,\mu;g) = \left(\int_{\mathbb{S}^{d_\theta-1}} W_k^k\big(\mathcal{H}_\mu(\cdot,\theta;g),\mathcal{H}_\mu(\cdot,\theta;g)\big)d\theta\right)^{\frac{1}{k}} = 0, \tag{37}$$

for $\forall \mu \in P_k(\mathbb{R}^d)$. Conversely, for $\forall \mu, \nu \in P_k(\mathbb{R}^d)$, if $\text{ASWD}_k(\mu,\nu;g)=0$, from the definition of the ASWD:

$$\text{ASWD}_k(\mu,\nu;g) = \left(\int_{\mathbb{S}^{d_\theta-1}} W_k^k\big(\mathcal{H}_\mu(\cdot,\theta;g),\mathcal{H}_\nu(\cdot,\theta;g)\big)d\theta\right)^{\frac{1}{k}} = 0, \tag{38}$$

Due to the non-negativity of $k$-th Wasserstein distance as it is a metric on $P_k(\mathbb{R}^d)$ and the continuity of $W_k(\cdot,\cdot)$ on $P_k(\mathbb{R}^d)$ (Villani, 2008), $W_k\big(\mathcal{H}_\mu(\cdot,\theta;g),\mathcal{H}_\nu(\cdot,\theta;g)\big)=0$ holds for $\forall \theta \in \mathbb{S}^{d_\theta-1}$ if and only if $\mathcal{H}_\mu(\cdot,\theta;g) \equiv \mathcal{H}_\nu(\cdot,\theta;g)$. Again, given the spatial Radon transform is injective when $g(\cdot)$ is injective (see the proof in Appendix A), $\mathcal{H}_\mu(\cdot,\theta;g) \equiv \mathcal{H}_\nu(\cdot,\theta;g)$ implies $\mu \equiv \nu$ if $g(\cdot)$ is injective.

In addition, if $g(\cdot)$ is not injective, the spatial Radon transform is not injective (see the proof in Appendix A), then $\exists \mu, \nu \in P_k(\mathbb{R}^d)$, $\mu \not\equiv \nu$ such that $\mathcal{H}_\mu(\cdot,\theta;g) \equiv \mathcal{H}_\nu(\cdot,\theta;g)$, which implies $\text{ASWD}_k(\mu,\nu;g)=0$ for $\mu \not\equiv \nu$. Therefore, the ASWD satisfies the identity of indiscernibles if and only if $g(\cdot)$ is injective.

**Non-negativity:** The three axioms of a distance metric, i.e. symmetry, triangle inequality, and identity of indiscernibles imply the non-negativity of the ASWD. Since the Wasserstein distance is non-negative,

for $\forall \mu, \nu \in P_k(\mathbb{R}^d)$, it can also be straightforwardly proved the ASWD between $\mu$ and $\nu$ is non-negative:

$$
\begin{aligned}
\mathrm{ASWD}_k(\mu,\nu;g) &= \left( \int_{\mathbb{S}^{d_\theta-1}} W_k^k \left( \mathcal{H}_\mu(\cdot,\theta;g), \mathcal{H}_\nu(\cdot,\theta;g) \right) d\theta \right)^{\frac{1}{k}} \\
&\geq \left( \int_{\mathbb{S}^{d_\theta-1}} 0^k d\theta \right)^{\frac{1}{k}} = 0.
\end{aligned}
\tag{39}
$$

Therefore, the ASWD is a metric on $P_k(\mathbb{R}^d)$ if and only if $g(\cdot)$ is injective. $\qquad\square$

## APPENDIX D    PROOF OF COROLLARY 1.1

We first introduce Lemma 2 to support the proof of Corollary 1.1.

**Lemma 2.** *For $\lambda \in (1, +\infty)$, the optimal mapping $g^*(\cdot)$ defined by Equation (17) satisfies $||g^*(x)||_2 < \infty$ for $\forall x \in \mathbb{R}^d \sim \mu$ and $\forall x \in \mathbb{R}^d \sim \nu$.*

*Proof.* Recall that in Equation (16) the ASWD can be rewritten as:

$$
\begin{aligned}
\text{ASWD}_k(\mu,\nu;g) &= \text{SWD}_k(\hat{\mu}_g,\hat{\nu}_g) \\
&= \left( \int_{\mathbb{S}^{d_\theta-1}} W_k^k \big( \mathcal{R}_{\hat{\mu}_g}(\cdot,\theta), \mathcal{R}_{\hat{\nu}_g}(\cdot,\theta) \big) d\theta \right)^{\frac{1}{k}},
\end{aligned}
\tag{40}
$$

where transformed variables $\hat{x} = g(x) \sim \hat{\mu}_g$ for $x \sim \mu$ and $\hat{y} = g(y) \sim \hat{\nu}_g$ for $y \sim \nu$, respectively. Combining the equation above with Equation (2):

$$
\begin{aligned}
\text{ASWD}_k(\mu,\nu;g) &= \left( \int_{\mathbb{S}^{d_\theta-1}} W_k^k \big( \mathcal{R}_{\hat{\mu}_g}(\cdot,\theta), \mathcal{R}_{\hat{\nu}_g}(\cdot,\theta) \big) d\theta \right)^{\frac{1}{k}} \\
&= \left( \int_{\mathbb{S}^{d_\theta-1}} \int_0^1 |F_{P_\theta \# \hat{\mu}_g}^{-1}(z) - F_{P_\theta \# \hat{\nu}_g}^{-1}(z)|^k dz d\theta \right)^{\frac{1}{k}} \\
&\leq \left( \int_{\mathbb{S}^{d_\theta-1}} \int_0^1 \big( |F_{R_\theta(\hat{\mu}_g)}^{-1}(z)| + |F_{R_\theta(\hat{\nu}_g)}^{-1}(z)| \big)^k dz d\theta \right)^{\frac{1}{k}},
\end{aligned}
\tag{41}
$$

where $\#$ denotes the push forward operator, $P_\theta : x \in \mathbb{R}^{d_\theta} \to \langle x, \theta \rangle \in \mathbb{R}$, and $R_\theta(\hat{\mu}_g) = P_\theta \# \hat{\mu}_g$, $R_\theta(\hat{\nu}_g) = P_\theta \# \hat{\nu}_g$ refer to one-dimensional measures obtained by slicing $\hat{\mu}_g$, $\hat{\nu}_g$ with a unit vector $\theta$, $F_{R_\theta(\hat{\mu}_g)}^{-1}$ and $F_{R_\theta(\hat{\nu}_g)}^{-1}$ are inverse cumulative distribution functions (CDFs) of $R_\theta(\hat{\mu}_g)$ and $R_\theta(\hat{\nu}_g)$, respectively.

By repeatedly applying the Minkowski's inequality to Equation (41), we obtain the following inequalities:

$$
\begin{aligned}
\text{ASWD}_k(\mu,\nu;g) &\leq \left( \int_{\mathbb{S}^{d_\theta-1}} \int_0^1 \big( |F_{R_\theta(\hat{\mu}_g)}^{-1}(z)| + |F_{R_\theta(\hat{\nu}_g)}^{-1}(z)| \big)^k dz d\theta \right)^{\frac{1}{k}} \\
&\leq \left( \int_{\mathbb{S}^{d_\theta-1}} \left[ \left( \int_0^1 |F_{R_\theta(\hat{\mu}_g)}^{-1}(z)|^k dz \right)^{\frac{1}{k}} + \left( \int_0^1 |F_{R_\theta(\hat{\nu}_g)}^{-1}(z)|^k dz \right)^{\frac{1}{k}} \right]^k d\theta \right)^{\frac{1}{k}}, \\
&\leq \left[ \int_{\mathbb{S}^{d_\theta-1}} \int_0^1 |F_{R_\theta(\hat{\mu}_g)}^{-1}(z)|^k dz d\theta \right]^{\frac{1}{k}} + \left[ \int_{\mathbb{S}^{d_\theta-1}} \int_0^1 |F_{R_\theta(\hat{\nu}_g)}^{-1}(z)|^k dz d\theta \right]^{\frac{1}{k}}.
\end{aligned}
\tag{42}
$$

Let $s = \langle \hat{x}, \theta \rangle$, then $z = F_{R_\theta(\hat{\mu}_g)}(s)$, $dz = dF_{R_\theta(\hat{\mu}_g)}(s)$:

$$
\begin{aligned}
\int_0^1 |F_{R_\theta(\hat{\mu}_g)}^{-1}(z)|^k dz &= \int_{\mathbb{R}} |s|^k dF_{R_\theta(\hat{\mu}_g)}(s) \\
&= \int_{\mathbb{R}^{d_\theta}} |\langle \hat{x}, \theta \rangle|^k d\hat{\mu}_g \\
&= \int_{\mathbb{R}^d} |\langle g(x), \theta \rangle|^k d\mu,
\end{aligned}
\tag{43}
$$

where the last two equations are obtained through the definitions of the push-forward operators. Therefore, the following inequalities hold:

$$
\begin{aligned}
\text{ASWD}_k(\mu,\nu;g) &\leq \left[ \int_{\mathbb{S}^{d_\theta-1}} \int_0^1 |F_{R_\theta(\hat{\mu}_g)}^{-1}(z)|^k dz d\theta \right]^{\frac{1}{k}} + \left[ \int_{\mathbb{S}^{d_\theta-1}} \int_0^1 |F_{R_\theta(\hat{\nu}_g)}^{-1}(z)|^k dz d\theta \right]^{\frac{1}{k}} \\
&= \left[ \int_{\mathbb{S}^{d_\theta-1}} \int_{\mathbb{R}^d} |\langle g(x), \theta \rangle|^k d\mu d\theta \right]^{\frac{1}{k}} + \left[ \int_{\mathbb{S}^{d_\theta-1}} \int_{\mathbb{R}^d} |\langle g(y), \theta \rangle|^k d\nu d\theta \right]^{\frac{1}{k}} \\
&\leq \mathbb{E}_{x \sim \mu}^{\frac{1}{k}} \big[ ||g(x)||_2^k \big] + \mathbb{E}_{y \sim \nu}^{\frac{1}{k}} \big[ ||g(y)||_2^k \big].
\end{aligned}
\tag{44}
$$

Then we obtain the following inequalities for the optimization objective:

$$
\begin{aligned}
&\mathrm{ASWD}_k(\mu,\nu;g) - L(\mu,\nu,\lambda;g) \\
&\leq \left(\mathbb{E}^{\frac{1}{k}}_{x\sim\mu}\left[||g(x)||_2^k\right] + \mathbb{E}^{\frac{1}{k}}_{y\sim\nu}\left[||g(y)||_2^k\right]\right) - \lambda\left(\mathbb{E}^{\frac{1}{k}}_{x\sim\mu}\left[||g(x)||_2^k\right] + \mathbb{E}^{\frac{1}{k}}_{y\sim\nu}\left[||g(y)||_2^k\right]\right) \\
&= (1-\lambda)\left(\mathbb{E}^{\frac{1}{k}}_{x\sim\mu}\left[||g(x)||_2^k\right] + \mathbb{E}^{\frac{1}{k}}_{y\sim\nu}\left[||g(y)||_2^k\right]\right).
\end{aligned}
\tag{45}
$$

When we set $\lambda\in(1,+\infty)$, if $\exists x\in\mathbb{R}^d\sim\mu$ or $y\in\mathbb{R}^d\sim\nu$ such that $||g(x)||_2\to\infty$ or $||g(y)||_2\to\infty$, the optimization objective approaches negative infinity, implying $g(\cdot)$ is not the optimal mapping. Therefore, by adopting Equation (17) as the optimization objective, the optimal mapping $g^*(\cdot)$ satisfies $||g^*(x)||_2<\infty$ for $\forall x\in\mathbb{R}^d\sim\mu$ and $\forall x\in\mathbb{R}^d\sim\nu$.

$\square$

**Remark 6.** *It is worth noting that $\lambda>1$ in Corollary 1.1 is a sufficient condition for the supremum of the optimization objective to be non-positive and $||g^*(x)||_2<\infty$ for $\forall x\in\mathbb{R}^d\sim\mu$ and $\forall x\in\mathbb{R}^d\sim\nu$. $0<\lambda\leq 1$ can also lead to finite $||g(x)||_2$ in various scenarios. Specifically, the upper bound of the optimization objective given in Equation (44) is obtained by applying:*

$$
|\langle g(x),\theta\rangle| = ||g(x)||_2|\cos(\alpha)| \leq ||g(x)||_2,
\tag{46}
$$

*where $\alpha$ is the angle between $\theta$ and $g(x)$. In high-dimensional spaces, Equation (46) gives a very loose bound since in high-dimensional spaces the majority of sampled $\theta$ would be nearly orthogonal to $g(x)$ and $\cos(\alpha)$ is nearly zero with high probability (Kolouri et al., 2019a). Empirically we found that across all the experiment results, $\lambda$ in a candidate set of $\{0.01,0.05,0.1,0.5,1,10,100\}$ all lead to finite $g^*(\cdot)$.*

We now prove Corollary 1.1, i.e $\mathrm{ASWD}_k(\mu,\nu;g^*)$ is a metric on $P_k(\mathbb{R}^d)$, where $g^*(\cdot)$ is the optimal mapping defined by Equation (17) for $\lambda\in(1,+\infty)$.

*Proof.* **Symmetry:** Since the $k$-Wasserstein distance is a metric thus symmetric (Villani, 2008):

$$
W_k\left(\mathcal{H}_\mu(\cdot,\theta;g^*),\mathcal{H}_\nu(\cdot,\theta;g^*)\right) = W_k\left(\mathcal{H}_\nu(\cdot,\theta;g^*),\mathcal{H}_\mu(\cdot,\theta;g^*)\right).
\tag{47}
$$

Therefore,

$$
\begin{aligned}
\mathrm{ASWD}_k(\mu,\nu;g^*) &= \left(\int_{\mathbb{S}^{d_\theta-1}} W_k^k\left(\mathcal{H}_\mu(\cdot,\theta;g^*),\mathcal{H}_\nu(\cdot,\theta;g^*)\right)d\theta\right)^{\frac{1}{k}} \\
&= \left(\int_{\mathbb{S}^{d_\theta-1}} W_k^k\left(\mathcal{H}_\nu(\cdot,\theta;g^*),\mathcal{H}_\mu(\cdot,\theta;g^*)\right)d\theta\right)^{\frac{1}{k}} = \mathrm{ASWD}_k(\nu,\mu;g^*).
\end{aligned}
$$

**Triangle inequality:** From Lemma 2, when $\lambda\in(1,+\infty)$, the optimal mapping $g^*(\cdot)$ satisfies $||g^*(x)||_2<\infty$ for $\forall x\in\mathbb{R}^d\sim\mu$ and $\forall x\in\mathbb{R}^d\sim\nu$, hence $\mathrm{ASWD}_k(\nu,\mu;g^*)$ is finite due to Equation (16). We then prove that $\mathrm{ASWD}_k(\nu,\mu;g^*)$ satisfies the triangle inequality.

Denote by $g_1^*$, $g_2^*$, and $g_3^*$ optimal mappings that result in the supremum of Equation (19) between $\mu_1$ and $\mu_2$, $\mu_1$ and $\mu_3$, $\mu_2$ and $\mu_3$, respectively, since $\mathrm{ASWD}_k(\mu_1,\mu_2;g_1^*)$, $\mathrm{ASWD}_k(\mu_1,\mu_3;g_2^*)$, and $\mathrm{ASWD}_k(\mu_2,\mu_3;g_3^*)$ are finite, the following equations hold:

$$
\mathrm{ASWD}_k(\mu_1,\mu_2;g_1^*) \leq \mathrm{ASWD}_k(\mu_1,\mu_3;g_1^*) + \mathrm{ASWD}_k(\mu_2,\mu_3;g_1^*)
\tag{48}
$$

$$
\leq \sup_g\{\mathrm{ASWD}_k(\mu_1,\mu_3;g)\} + \sup_g\{\mathrm{ASWD}_k(\mu_2,\mu_3;g)\}
\tag{49}
$$

$$
= \mathrm{ASWD}_k(\mu_1,\mu_3;g_2^*) + \mathrm{ASWD}_k(\mu_2,\mu_3;g_3^*),
\tag{50}
$$

where the first inequality are from the metric property of the ASWD.

**Identity of indiscernibles:** Since $W_k(\mu,\mu)=0$ for $\forall\mu\in P_k(\mathbb{R}^d)$, we have

$$
\mathrm{ASWD}_k(\mu,\mu;g^*) = \left(\int_{\mathbb{S}^{d_\theta-1}} W_k^k\left(\mathcal{H}_\mu(\cdot,\theta;g^*),\mathcal{H}_\mu(\cdot,\theta;g^*)\right)d\theta\right)^{\frac{1}{k}} = 0,
\tag{51}
$$

for $\forall \mu \in P_k(\mathbb{R}^d)$. Conversely, for $\forall \mu, \nu \in P_k(\mathbb{R}^d)$, if $\text{ASWD}_k(\mu, \nu; g^*) = 0$, from Equation (15):

$$\text{ASWD}_k(\mu, \nu; g^*) = \left( \int_{\mathbb{S}^{d_\theta - 1}} W_k^k \big( \mathcal{H}_\mu(\cdot, \theta; g^*), \mathcal{H}_\nu(\cdot, \theta; g^*) \big) d\theta \right)^{\frac{1}{k}} = 0. \tag{52}$$

Due to the non-negativity of $k$-th Wasserstein distance as it is a metric on $P_k(\mathbb{R}^d)$ and the continuity of $W_k(\cdot, \cdot)$ on $P_k(\mathbb{R}^d)$ (Villani, 2008), $W_k \big( \mathcal{H}_\mu(\cdot, \theta; g^*), \mathcal{H}_\nu(\cdot, \theta; g^*) \big) = 0$ holds for $\forall \theta \in \mathbb{S}^{d_\theta - 1}$, which implies $\mathcal{H}_\mu(\cdot, \theta; g^*) \equiv \mathcal{H}_\nu(\cdot, \theta; g^*)$ for $\forall \theta \in \mathbb{S}^{d_\theta - 1}$. Therefore, given the spatial Radon transform is injective when $g^*(\cdot)$ is injective, $\mathcal{H}_\mu(\cdot, \theta; g^*) \equiv \mathcal{H}_\nu(\cdot, \theta; g^*)$ implies $\mu \equiv \nu$.

**Non-negativity:** Since the Wasserstein distance is non-negative, for $\forall \mu, \nu \in P_k(\mathbb{R}^d)$, the ASWD defined with optimal mappings $g(\cdot)$ between $\mu$ and $\nu$ is also non-negative:

$$\begin{aligned}
\text{ASWD}_k(\mu, \nu; g^*) &= \left( \int_{\mathbb{S}^{d_\theta - 1}} W_k^k \big( \mathcal{H}_\mu(\cdot, \theta; g^*), \mathcal{H}_\nu(\cdot, \theta; g^*) \big) d\theta \right)^{\frac{1}{k}} \\
&\geq \left( \int_{\mathbb{S}^{d_\theta - 1}} 0^k d\theta \right)^{\frac{1}{k}} = 0.
\end{aligned} \tag{53}$$

Therefore, the ASWD defined with the optimal mapping $g^*(\cdot)$ is non-negative, symmetric, and satisfies the triangle inequality and the identity of indiscernibles, i.e. the ASWD defined with optimal mappings $g^*(\cdot)$ is also a metric.

$\square$

## APPENDIX E   PSEUDOCODE FOR THE EMPIRICAL VERSION OF THE ASWD

---

**Algorithm 1** The augmented sliced Wasserstein distance. All of the for loops can be parallelized.

---

**Require:** Sets of samples $\{x_n \in \mathbb{R}^d\}_{n=1}^N$, $\{y_n \in \mathbb{R}^d\}_{n=1}^N$;
**Require:** Randomly initialized injective neural network $g_\omega(\cdot): \mathbb{R}^d \to \mathbb{R}^{d_\theta}$;
**Require:** Number of projections $L$, hyperparameter $\lambda$, learning rate $\epsilon$, number of iterations $M$;
1: Initialize $D=0, L_\lambda=0, m=1$;
2: **while** $\omega$ has not converged and $m \leq M$ **do**
3:     Draw a set of samples $\{\theta_l\}_{l=1}^L$ from $\in \mathbb{S}^{d_\theta-1}$;
4:     **for** $n=1$ to $N$ **do**
5:         Compute $g_\omega(x_n)$ and $g_\omega(y_n)$;
6:         Calculate the regularization term $L_\lambda \leftarrow L_\lambda + \lambda\left[\left(\frac{||g_\omega(x_n)||_2^k}{N}\right)^{\frac{1}{k}} + \left(\frac{||g_\omega(y_n)||_2^k}{N}\right)^{\frac{1}{k}}\right]$;
7:     **end for**
8:     **for** $l=1$ to $L$ **do**
9:         Compute $\beta(x_n,\theta_l) = \langle g_\omega(x_n),\theta_l\rangle$, $\beta(y_n,\theta_l) = \langle g_\omega(y_n),\theta_l\rangle$ for each $n$;
10:         Sort $\beta(x_n,\theta_l)$ and $\beta(y_n,\theta_l)$ in ascending order s.t. $\beta(x_{I_x^l[n]},\theta_l) \leq \beta(x_{I_x^l[n+1]},\theta_l)$ and $\beta(y_{I_y^l[n]},\theta_l) \leq \beta(y_{I_y^l[n+1]},\theta_l)$;
11:         Calculate the ASWD: $D \leftarrow D + \left(\frac{1}{L}\sum_{n=1}^N |\beta(x_{I_x^l[n]},\theta_l) - \beta(y_{I_y^l[n]},\theta_l)|^k\right)^{\frac{1}{k}}$;
12:     **end for**
13:     $\mathcal{L} \leftarrow D - L_\lambda$;
14:     Update $\omega$ by gradient ascent $\omega \leftarrow \omega + \epsilon \cdot \nabla_\omega \mathcal{L}$;
15:     Reset $D=0, L_\lambda=0$, update $m \leftarrow m+1$;
16: **end while**
17: Draw a set of samples $\{\theta_l\}_{l=1}^L$ from $\in \mathbb{S}^{d_\theta-1}$;
18: **for** $n=1$ to $N$ **do**
19:     Compute $g_\omega(x_n)$ and $g_\omega(y_n)$;
20: **end for**
21: **for** $l=1$ to $L$ **do**
22:     Compute $\beta(x_n,\theta_l) = \langle g_\omega(x_n),\theta_l\rangle$, $\beta(y_n,\theta_l) = \langle g_\omega(y_n),\theta_l\rangle$ for each $n$;
23:     Sort $\beta(x_n,\theta_l)$ and $\beta(y_n,\theta_l)$ in ascending order s.t. $\beta(x_{I_x^l[n]},\theta_l) \leq \beta(x_{I_x^l[n+1]},\theta_l)$ and $\beta(y_{I_y^l[n]},\theta_l) \leq \beta(y_{I_y^l[n+1]},\theta_l)$;
24:     Calculate the ASWD: $D \leftarrow D + \left(\frac{1}{L}\sum_{n=1}^N |\beta(x_{I_x^l[n]},\theta_l) - \beta(y_{I_y^l[n]},\theta_l)|^k\right)^{\frac{1}{k}}$;
25: **end for**
26: **Output:** Augmented sliced Wasserstein distance $D$.

---

In Algorithm 1, Equation (17) is used as the optimization objective, where the regularization term $L(\mu,\nu,\lambda;g_\omega) = \lambda(\mathbb{E}_{x\sim\mu}^{\frac{1}{k}}\left[||g_\omega(x)||_2^k\right] + \mathbb{E}_{y\sim\nu}^{\frac{1}{k}}\left[||g_\omega(y)||_2^k\right])$ is used. This particular choice of the regularization term facilitates the proofs to Lemma 2 and subsequently to Corollary 1.1 with details in Appendix D. In fact, we have also examined other types of regularization terms such as the $L_2$ norm of the output of $g(\cdot)$, and empirically they produce similar numerical results as the current regularization term.

## APPENDIX F    EXPERIMENTAL SETUPS

### F.1    HYPERPARAMETERS IN THE SLICED WASSERSTEIN FLOW EXPERIMENT

We randomly generate 500 samples both for target distributions and source distributions. We initialize the source distributions $\mu_0$ as standard normal distributions $\mathcal{N}(0, I)$, where $I$ is a 2-dimensional identity matrix. We update source distributions using Adam optimizer, and set the learning rate=0.002. For all methods, we set the order $k = 2$. When testing the ASWD, the number of iterations $M$ in Algorithm 1 is set to 10.

In the sliced Wasserstein flow experiment the mapping $g^*(\cdot)$ optimized by maximizing Equation (17) was found to be finite for all values of $\lambda$ in the set of {0.01, 0.05, 0.1, 0.5, 1, 10, 100}, which is a sufficient condition for the ASWD to be a valid metric as shown in the proof of corollary 1.1 provided in Appendix D. In addition, numerical results presented in Appendix G.2 indicate that empirical errors in the experiment are not sensitive to the choice of $\lambda$ in the candidate set {0.01, 0.05, 0.1, 0.5}. The reported results in the main paper are produced with $\lambda = 0.1$.

### F.2    NETWORK ARCHITECTURE IN THE GENERATIVE MODELING EXPERIMENT

Denote a convolutional layer whose kernel size is $s$ with $C$ kernels by $\mathrm{Conv}_C(s \times s)$, and a fully-connected layer whose input and output layer have $s_1$ and $s_2$ neurons by $\mathrm{FC}(s_1 \times s_2)$. The network structure used in the generative modeling experiment is configured to be the same as described in (Nguyen et al., 2021):

$$h_\psi : (64 \times 64 \times 3) \to \mathrm{Conv}_{64}(4 \times 4) \to \mathrm{LeakyReLU}(0.2) \to$$
$$\mathrm{Conv}_{128}(4 \times 4) \to \mathrm{BatchNormalization} \to \mathrm{LeakyReLU}(0.2) \to$$
$$\mathrm{Conv}_{256}(4 \times 4) \to \mathrm{BatchNormalization} \to \mathrm{LeakyReLU}(0.2) \to$$
$$\mathrm{Conv}_{512}(4 \times 4) \to \mathrm{BatchNormalization} \to \mathrm{Tanh} \xrightarrow{\mathrm{Output}} (512 \times 4 \times 4)$$

$$D_\Psi : \mathrm{Conv}_1(4 \times 4) \to \mathrm{Sigmoid} \xrightarrow{\mathrm{Output}} (1 \times 1 \times 1)$$

$$G_\Phi : z \in \mathbb{R}^{100} \to \mathrm{ConvTranspose}_{512}(4 \times 4) \to$$
$$\mathrm{BatchNormalization} \to \mathrm{ReLU} \to \mathrm{ConvTranspose}_{256}(4 \times 4) \to$$
$$\mathrm{BatchNormalization} \to \mathrm{ReLU} \to \mathrm{ConvTranspose}_{128}(4 \times 4) \to$$
$$\mathrm{BatchNormalization} \to \mathrm{ReLU} \to \mathrm{ConvTranspose}_{64}(4 \times 4) \to$$
$$\mathrm{BatchNormalization} \to \mathrm{ConvTranspose}_3(4 \times 4) \to \mathrm{Tanh}$$
$$\xrightarrow{\mathrm{Ouput}} (64 \times 64 \times 3)$$

$$\phi : \mathrm{FC}(8192 \times 8192) \xrightarrow{\mathrm{Output}} (8192)\text{-dimensional vector}$$

We train the models with the Adam optimizer, and set the batch size to 512. Following the setup in (Nguyen et al., 2021), the learning rate is set to 0.0005 and beta=(0.5, 0.999) for both CIFAR10 dataset and CelebA dataset. For all methods, we set the order $k$ to 2. For the ASWD, the number of iterations $M$ in Algorithm 1 is set to 5. The hyperparameter $\lambda$ is set to 1.01 to guarantee that the ASWD being a valid metric and introduce slightly larger regularization of the optimization objective due to the small output values from the feature layer $h_\psi$.

# APPENDIX G    ADDITIONAL
## RESULTS IN THE SLICED WASSERSTEIN FLOW EXPERIMENT

### G.1    FULL EXPERIMENTAL RESULTS ON THE SLICED WASSERSTEIN EXPERIMENT

Figure 4 shows the full experimental results on the sliced Wasserstein flow experiment.

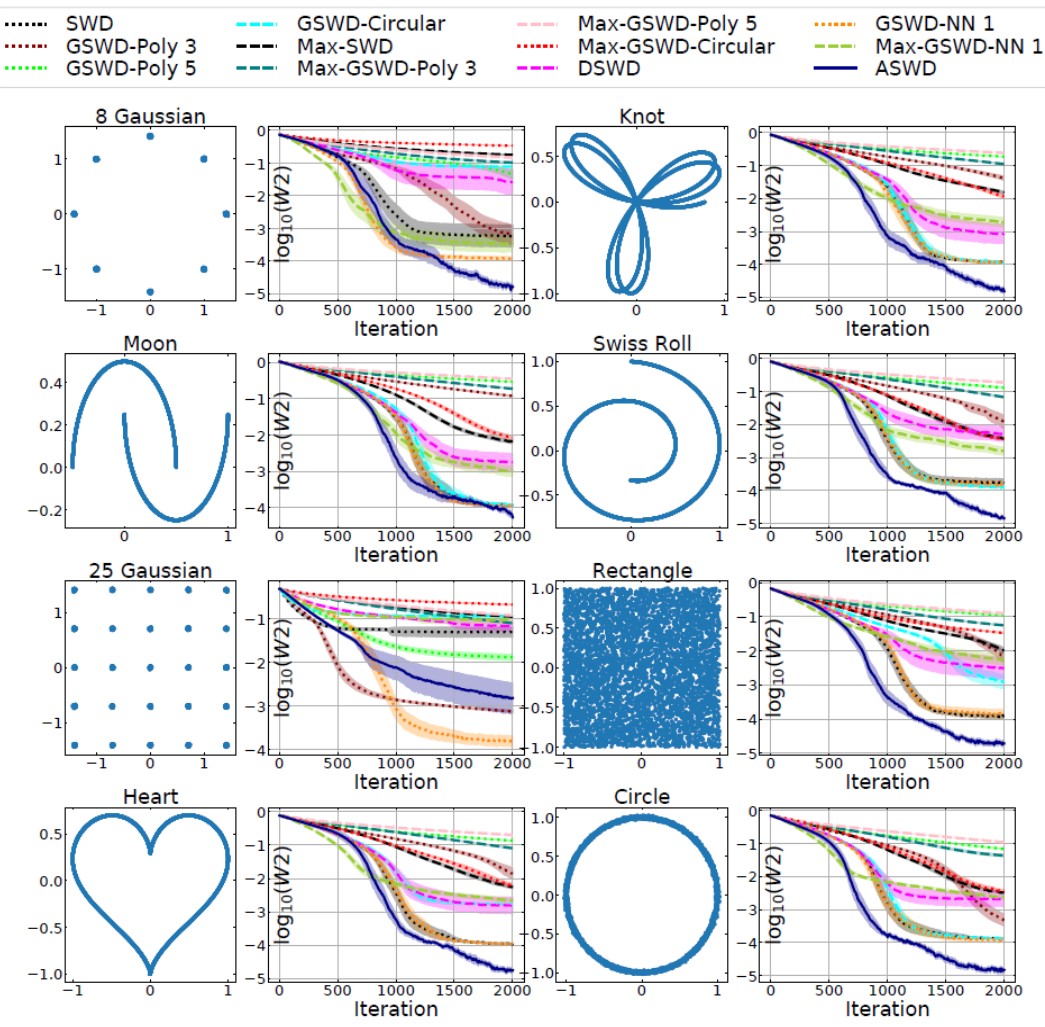

Figure 4: Full experimental results on the sliced Wasserstein flow example. The first and third columns are target distributions. The second and fourth columns are log 2-Wasserstein distances between the target distributions and the source distributions. The horizontal axis shows the number of training iterations. Solid lines and shaded areas represent the average values and 95% confidence intervals of log 2-Wasserstein distances over 50 runs.

## G.2 Ablation Study

**Impact of the injectivity and optimization of the mapping**

In this ablation study, we compare ASWDs constructed by different mappings to GSWDs with different predefined defining functions, and investigate the effects of the optimization and injectivity of the adopted mapping $g_\omega(\cdot)$ used in the ASWDs. In what follows, "ASWD-vanilla" is used to denote ASWDs that employ randomly initialized neural network $\phi_\omega(\cdot)$ to parameterize the injective mapping $g_\omega(\cdot) = [\cdot, \phi_\omega(\cdot)]$, i.e. the mapping $g_\omega(\cdot)$ is not optimized in the ASWD-vanilla and the results of ASWD-vanilla reported in Figure 5 are obtained by slicing with random hypersurfaces. Furthermore, the "ASWD-non-injective" refers to ASWDs that do not use the injectivity trick, i.e. the mapping $g_\omega(\cdot) = \phi_\omega(\cdot)$ is not guaranteed to be injective. In addition, the "ASWD-vanilla-non-injective" adopts both setups in the "ASWD-vanilla" and "ASWD-non-injective", resulting in a random non-injective mapping $g_\omega(\cdot)$. The reported experiment results in this ablation study is calculated over 50 runs, and the neural network $\phi_\omega(\cdot)$ is reinitialized randomly in each run.

From Figure 5, it can be observed that the ASWD-vanilla shows comparable performance to GSWDs defined by polynomial and circular defining functions, which implies GSWDs with predefined defining functions are as uninformative as slicing distributions with random hypersurfaces constructed by the ASWD. In GSWDs, the hypersurfaces are predefined and cannot be optimized since they are determined by the functional forms of the defining functions. On the contrary, we found that the optimization of hypersurfaces in the ASWD framework can help improve the performance of the slice-based Wasserstein distance. As in Figure 5, the ASWD and the ASWD-non-injective present significantly better performance than methods that do not optimize their hypersurfaces (ASWD-vanilla, ASWD-vanilla-non-injective, and GSWDs). In terms of the impact of the injectivity of the mapping $g_\omega$, in this experiment, the ASWD-vanilla exhibits smaller 2-Wasserstein distances than the ASWD-vanilla-non-injective in all tested distributions, and the ASWD leads to more stable training than the ASWD-non-injective. Therefore, the injectivity of the mapping $g_\omega(\cdot)$ does not only guarantee the ASWD to be a valid distance metric as proved in Section 3, but also better empirical performance in this experiment setup.

**Impact of the regularization coefficient**

We also evaluated the sensitivity of performance of the ASWD with respect to the regularization coefficient $\lambda$. The ASWD is evaluated with different values of $\lambda$ and compared with other slice-based Wasserstein metrics in this ablation study. The numerical results presented in Figure 6 indicates that different values of $\lambda$ in the candidate set $\{0.01, 0.05, 0.1, 0.5\}$ lead similar performance of the ASWD, i.e the performance of the ASWD is not sensitive to $\lambda$. Additionally, the ASWDs with different values of $\lambda$ in the candidate set outperform the other evaluated slice-based Wasserstein metrics.

We have also evaluated the performance of the ASWD when the range of $\lambda$ is much larger than in the candidate set. Specifically, as presented in Figure 7, when $\lambda$ is set to be large values, e.g 10 or 100, the resulted ASWD leads to decreased performance on par with the SWD. This is consistent with our expectation that excessive regularization will eliminate nonlinearity as discussed in Remark 2, leading to similar performance with the SWD.

In addition, the effect of the regularization term on the performance of Max-GSWD-NN was also investigated in this ablation study. The performance of the Max-GSWD-NN and the Max-GSWD-NN trained with the regularization term used in the ASWD are compared in Figure 8. From the numerical results presented in Figure 8, the Max-GSWD-NN 1 with regularization leads to performance similar to the Max-GSWD-NN 1 without regularization, implying that the performance gap between the ASWD and the Max-GSWD-NN is not due to the introduction of the regularization term.

**Choice of injective mapping**

We reported in Figure 9 the performance of the ASWD defined with other types of injective mappings other than Equation (18). In particular, we examined two invertible mappings, including the planar flow and radial flow (Rezende and Mohamed, 2015), as alternatives to the injective mapping defined by Equation (18). The numerical results presented in Figure 9 show that the ASWD defined with planar flow and radial flow produced better performance than GSWD variants in most setups. They exhibit slightly worse performance compared with the ASWD with injective mapping defined in Equation (18), possibly due to the additional restriction in invertible mapping imposed by the planar flow and radial flow.

**Choice of the dimensionality $d_\theta$ of the augmented space**

To investigate how the dimensionality $d_\theta$ of the augmented space affects the performance of the ASWD, different choices of $d_\theta$ are employed in the ASWD. Specifically, the injective network architecture $g_\omega(x) = [x, \phi_\omega(x)] : \mathbb{R}^d \to \mathbb{R}^{d_\theta}$ given in Equation (18) is adopted and $\phi_\omega$ is set to be single fully-connected neural networks whose output dimension equals $\{1, 2, 3, 4\}$ times its input dimension, i.e $d_\theta = \{2d, 3d, 4d, 5d\}$, respectively, where $d$ is the dimensionality of $x$. The numerical results are presented in Figure 10, and it can be found that the ASWDs present similar results across different choices of $d_\theta$. It can also be observed in Figure 10 that the ASWDs with different choices of $d_\theta$ consistently produce better performance than the other evaluated slice-based Wasserstein metrics.

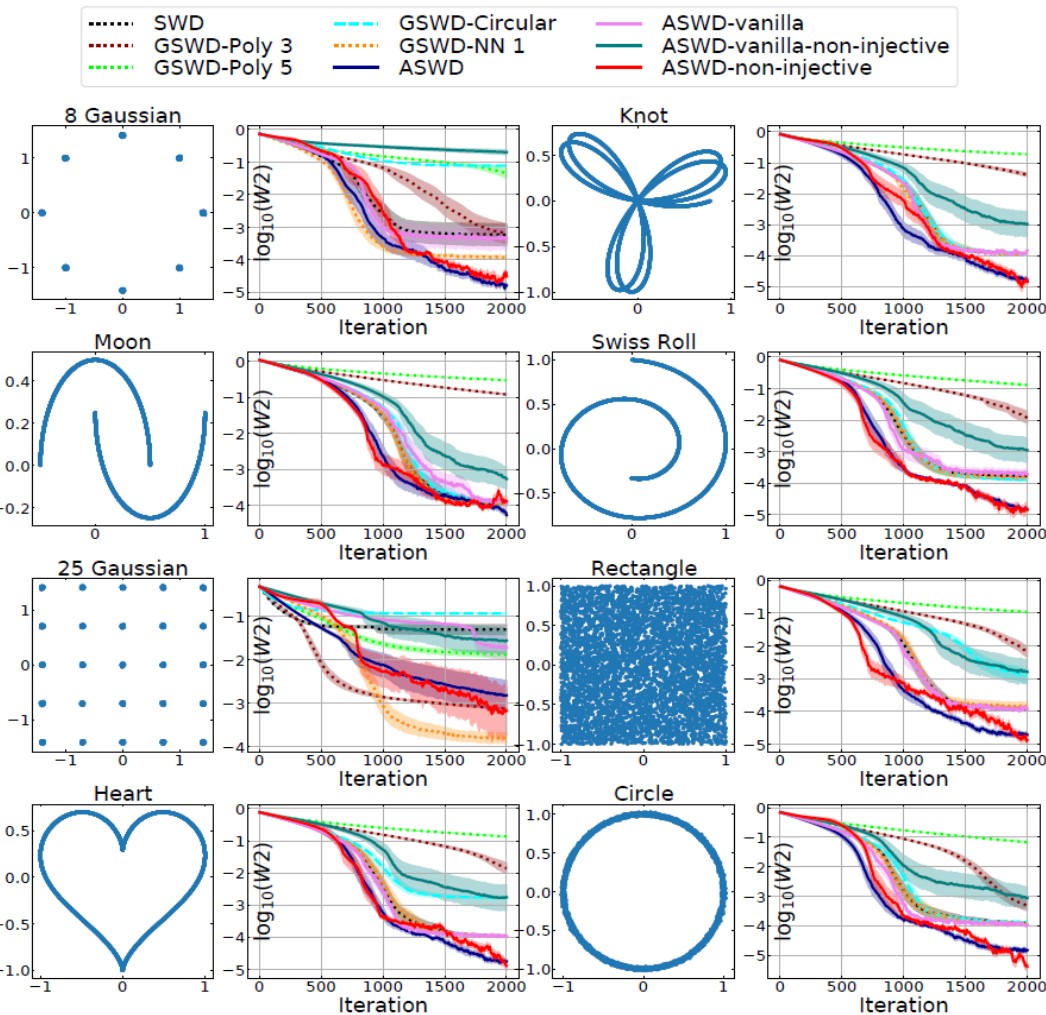

Figure 5: Ablation study on the impact from injective neural networks and the optimization of hypersurfaces on the ASWD. ASWDs with different mappings are compared to GSWDs with different defining functions. The first and third columns show target distributions. The second and fourth columns plot log 2-Wasserstein distances between the target distributions and the source distributions. In the second and fourth columns, the horizontal axis shows the number of training iterations. Solid lines and shaded areas represent the average values and 95% confidence intervals of log 2-Wasserstein distances over 50 runs.

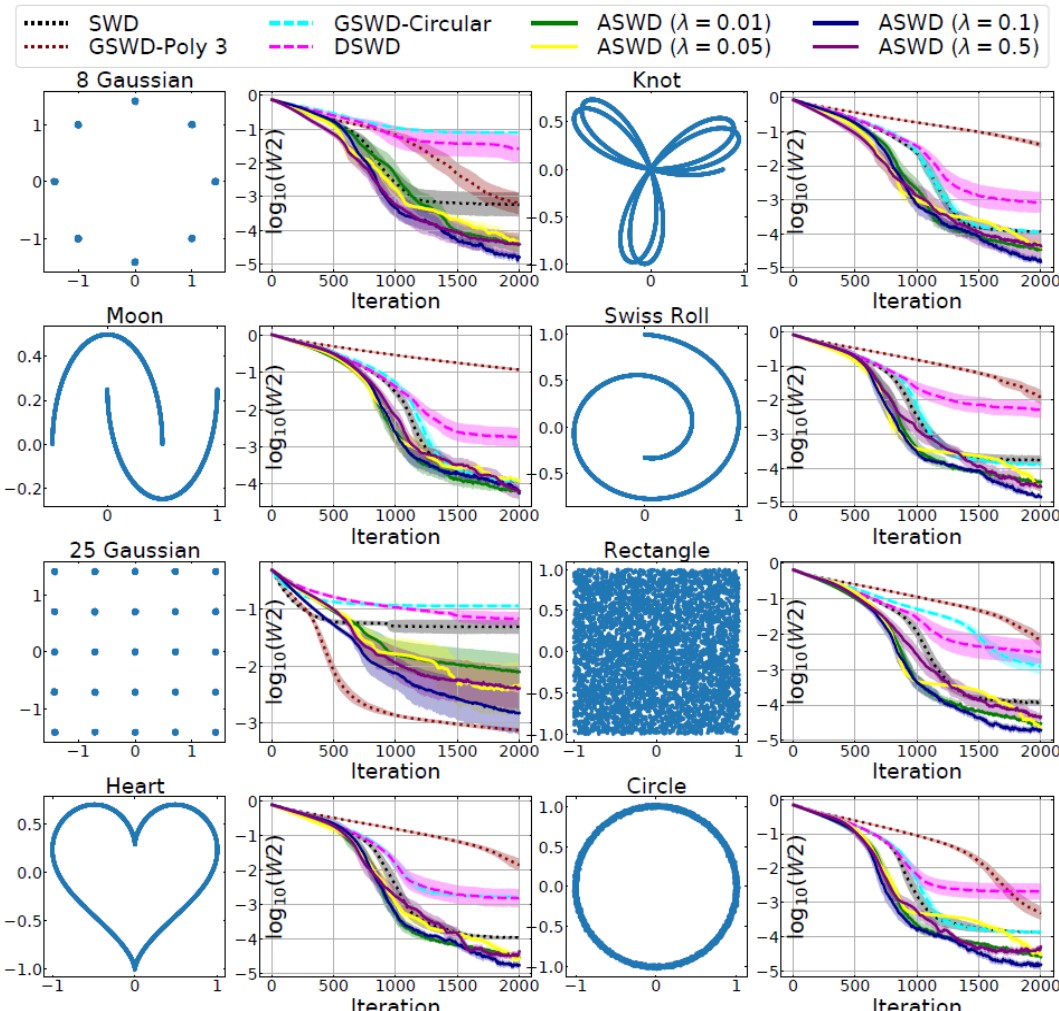

Figure 6: Ablation study on the impact of the regularization coefficient $\lambda$. The performance of the ASWDs with different values of $\lambda$ are compared with other slice-based Wasserstein metrics. The first and third columns show target distributions. The second and fourth columns plot log 2-Wasserstein distances between the target distributions and the source distributions. In the second and fourth columns, the horizontal axis shows the number of training iterations. Solid lines and shaded areas represent the average values and 95% confidence intervals of log 2-Wasserstein distances over 50 runs.

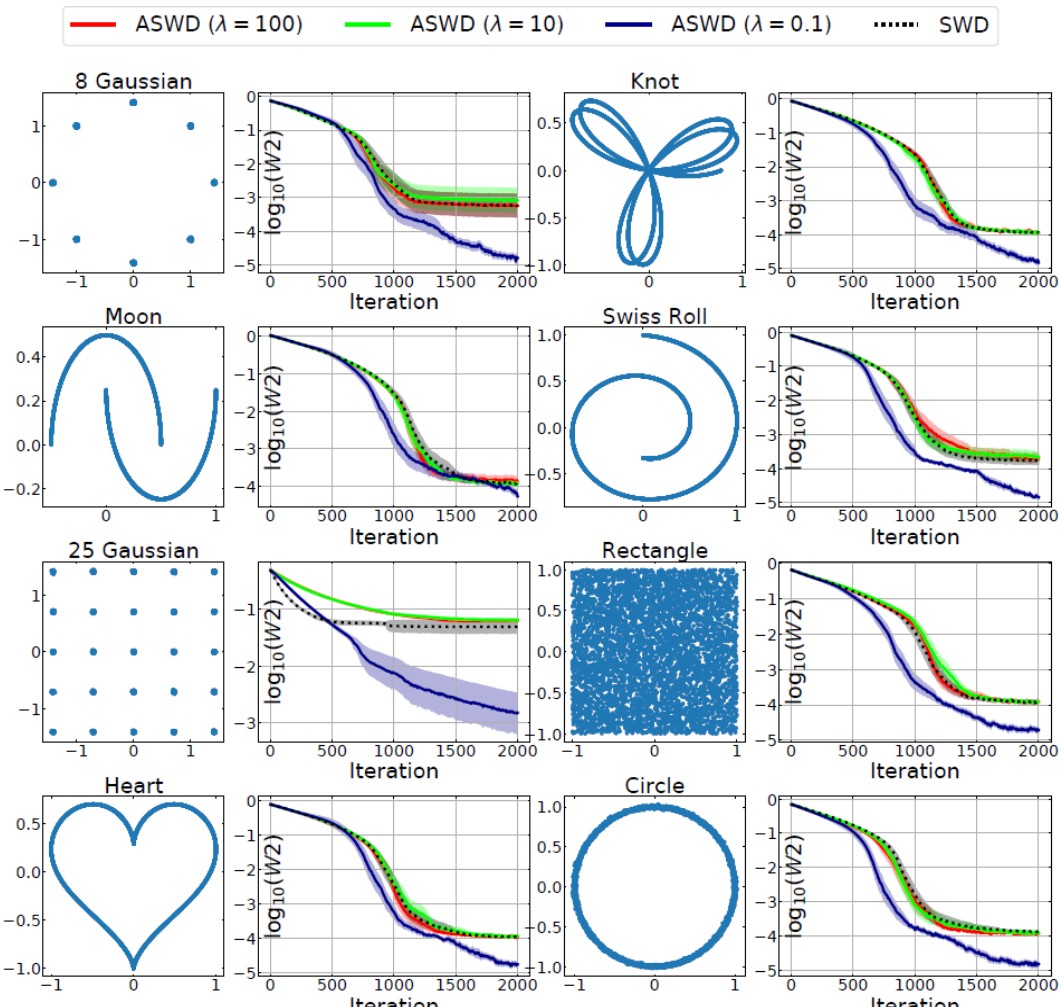

Figure 7: Ablation study on the impact from large values of $\lambda$. The performance of the ASWDs with large values of $\lambda$, e.g 10 and 100, are compared with the SWD. The first and third columns show target distributions. The second and fourth columns plot log 2-Wasserstein distances between the target distributions and the source distributions. In the second and fourth columns, the horizontal axis shows the number of training iterations. Solid lines and shaded areas represent the average values and 95% confidence intervals of log 2-Wasserstein distances over 50 runs.

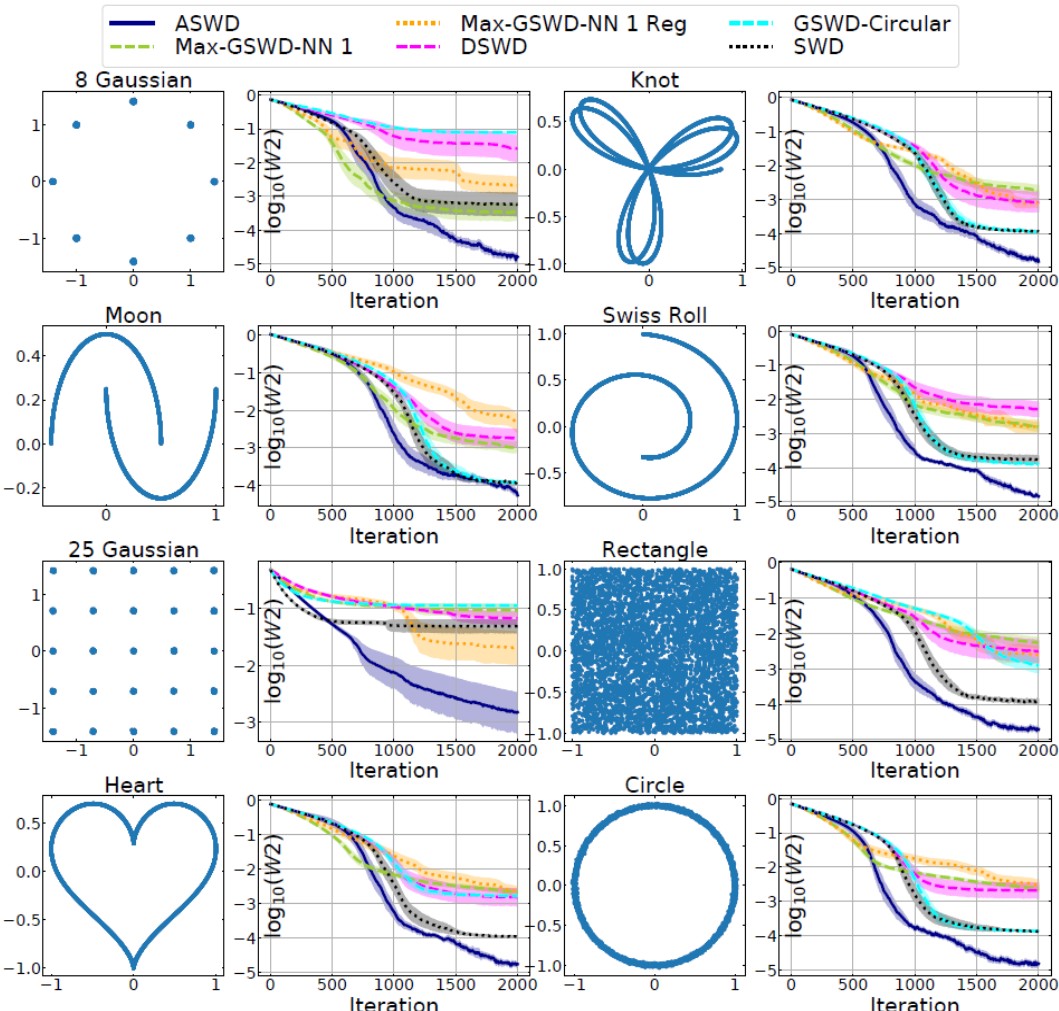

Figure 8: Ablation study on the impact from the regularization term on the performance of the Max-GSW-NN. The first and third columns show target distributions. The second and fourth columns plot log 2-Wasserstein distances between the target distributions and the source distributions. In the second and fourth columns, the horizontal axis shows the number of training iterations. Solid lines and shaded areas represent the average values and 95% confidence intervals of log 2-Wasserstein distances over 50 runs.

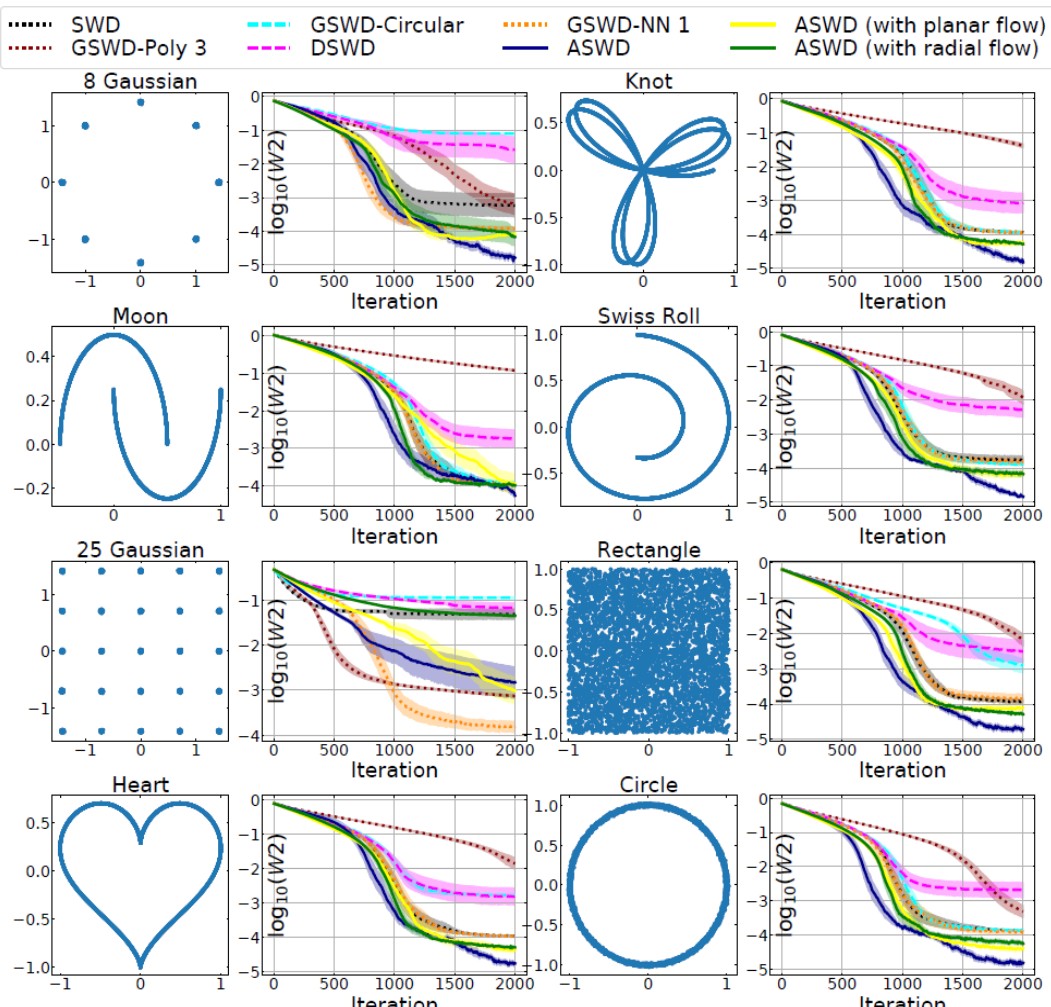

Figure 9: Ablation study on the impact from the choice of injective networks. The performance of the ASWDs with different types of injective networks are compared with other slice-based Wasserstein metrics. The first and third columns show target distributions. The second and fourth columns plot log 2-Wasserstein distances between the target distributions and the source distributions. In the second and fourth columns, the horizontal axis shows the number of training iterations. Solid lines and shaded areas represent the average values and 95% confidence intervals of log 2-Wasserstein distances over 50 runs.

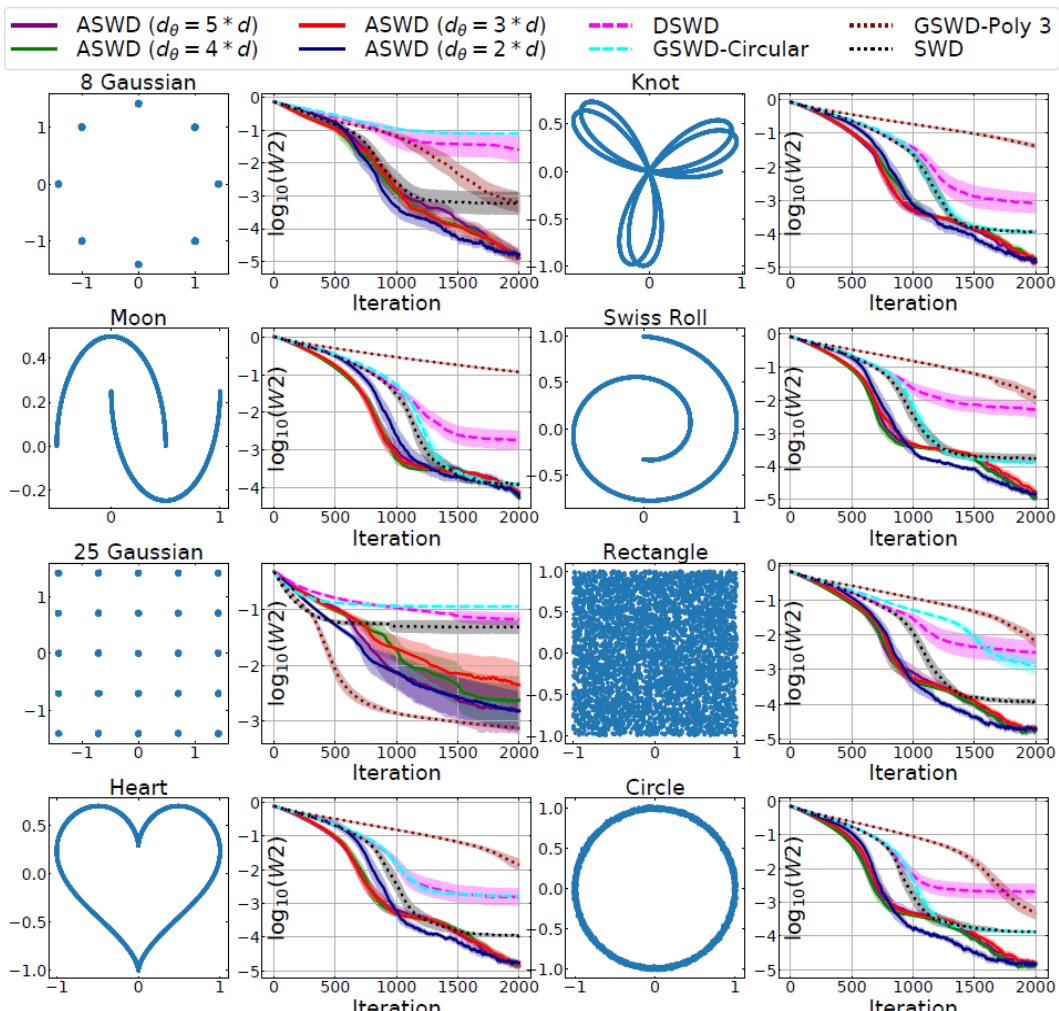

Figure 10: Ablation study on the choice of the dimensionality $d_\theta$ of the augmented space. The performance of the ASWDs with different choices of $d_\theta$ are compared with other slice-based Wasserstein metrics. The first and third columns show target distributions. The second and fourth columns plot log 2-Wasserstein distances between the target distributions and the source distributions. In the second and fourth columns, the horizontal axis shows the number of training iterations. Solid lines and shaded areas represent the average values and 95% confidence intervals of log 2-Wasserstein distances over 50 runs.

## APPENDIX H    ADDITIONAL
### RESULTS IN THE GENERATIVE MODELING EXPERIMENT

### H.1    SAMPLES OF GENERATED IMAGES OF CIFAR10 AND CELEBA DATASETS

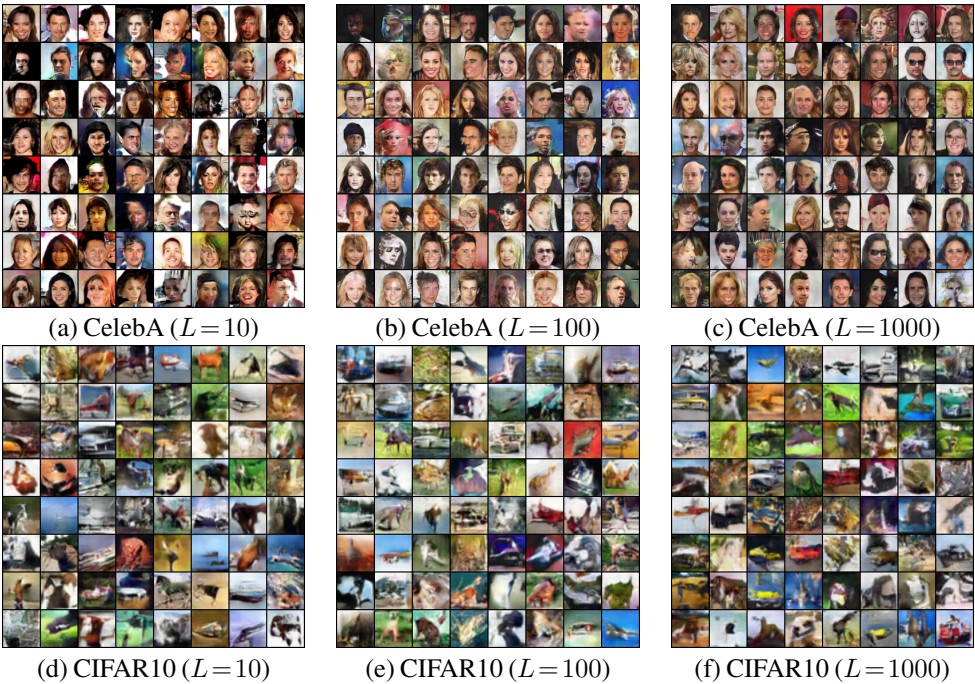

|   |   |   |
|---|---|---|
| (a) CelebA ($L=10$) | (b) CelebA ($L=100$) | (c) CelebA ($L=1000$) |
| (d) CIFAR10 ($L=10$) | (e) CIFAR10 ($L=100$) | (f) CIFAR10 ($L=1000$) |

Figure 11: Visualized experimental results of the ASWD on CelebA and CIFAR10 dataset with 10, 100, 1000 projections. The first row shows randomly selected samples of generated CelebA images, the second row shows randomly selected samples of generated CIFAR10 images.

### H.2    EXPERIMENT RESULTS ON MNIST DATASET

In the generative modelling experiment on the MNIST dataset, we train a generator by minimizing different slice-based Wasserstein metrics, including the ASWD, the DSWD, the GSWD (circular), and the SWD. Denote by $G_\Phi$ the generator, the training objective of the experiment can be formulated as (Bernton et al., 2019):

$$\min_\Phi \mathbb{E}_{x\sim p_r, z\sim p_z}[\text{SWD}(x, G_\Phi(z))], \tag{54}$$

where $p_z$ and $p_r$ are the prior of latent variable $z$ and the real data distribution, respectively. In other words, the SWD, or other slice-based Wasserstein metrics, can be considered as a discriminator in this framework. By replacing the SWD with the ASWD, the DSWD, and the GSWD, we compare the performance of learned generative models trained with different metrics. In this experiment, different methods are compared using different number of projections $L=\{10, 1000\}$. The 2-Wasserstein distance and the SWD between generated images and real images are used as metrics for evaluating performances of different generative models. The experiment results are presented in Figure 12. **Quality of generated images and convergence rate:** It can be observed from Figure 12 that the ASWD outperforms all the other methods regarding both the 2-Wasserstein distance and the SWD between generated and real images. In particular, the generative model trained with the ASWD produces smaller 2-Wasserstein distances within less iteration, which implies the generated images are of higher quality and the ASWD leads to higher convergence rates of generative models. In addition, the ASWD shows that it is able to generate higher quality images than the SWD and the GSWD with 1000 projections using only as less

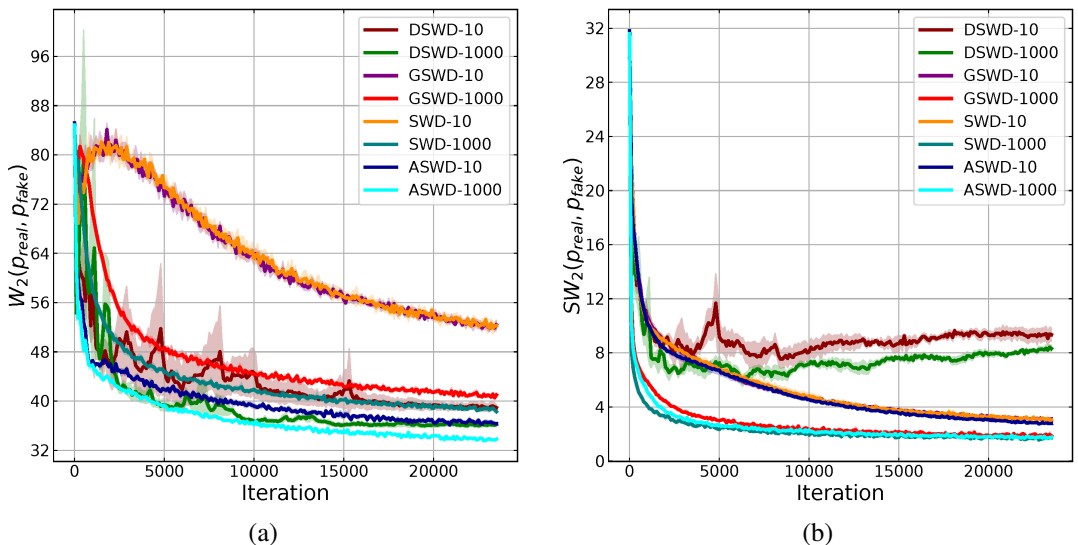

(a)                    (b)

Figure 12: Visualized experimental results of different slice-based Wasserstein metrics on the MNIST dataset with 10, 1000 projections. (a) Comparison between the SWD, the GSWD, the DSWD, and the ASWD using the 2-Wasserstein distance between fake and real images as the evaluation metric. (b) Comparison between the SWD, the GSWD, the DSWD, and the ASWD using the SWD between fake and real images as the evaluation metric.

as 10 projections. In other words, the ASWD has higher projection efficiency than the other slice-based Wasserstein metrics. The ASWD also has the smallest SWD distance as shown in Figure 12. Although the SWD converges slightly faster than the ASWD in terms of the SWD between fake and real images, this is due to the training objective and the evaluated metric are the same for the SWD. Randomly selected images generated by different slice-based Wasserstein metrics are presented in Figure 13.

**Computation cost of the ASWD:** The execution time per mini-batch (512 samples) of different methods are compared in Figure 14a. We evaluate the SWD by varying the number of projections in the set $\{10, 1000, 10000\}$ and all the other methods in the set $\{10, 1000\}$. We found that although the SWD requires much fewer computational time than the DSWD and the ASWD, the quality of generated data is poor even when the number of projections $L$ increases to 10000. The GSWD is also computationally efficient when using a 10 projections, but it requires the highest execution time and generates the highest 2-Wasserstein distance among all compared methods when the number of projections increases to 1000. The huge difference in the execution time of the GSWD with 10 and 1000 projections is due to the GSWD needs to calculate distance matrices of shape $N \times L$, where $N$ and $L$ are the number of samples and projections respectively, which is more computationally expensive than calculating inner products when the number of projections $L$ increases. The DSWD requires a similar computational time as the ASWD in this example, while the ASWD generates higher quality images in terms of 2-Wasserstein distances.

Besides, we have also evaluated the effect of batch size on the computation cost of different slice-based Wasserstein metrics. Due to the out-of-memory error caused by the excessively high computation cost of the GSWDs in high-dimensional space, the GSWD is not included in this comparison. Specifically, the ASWD, the DSWD, and the SWD are compared in this experiment, and the computation time of the evaluated methods with $L = 10, 1000$ projections and $N = \{2^{13}, 2^{14}, 2^{15}, 2^{16}\}$ samples are reported in Figure 14b. From the results presented in Figure 14b, it can be observed that, similar to the computational complexity of the DSWD and the SWD, the computational complexity of the ASWD empirically tends to scale in $\mathcal{O}(N \log N)$.

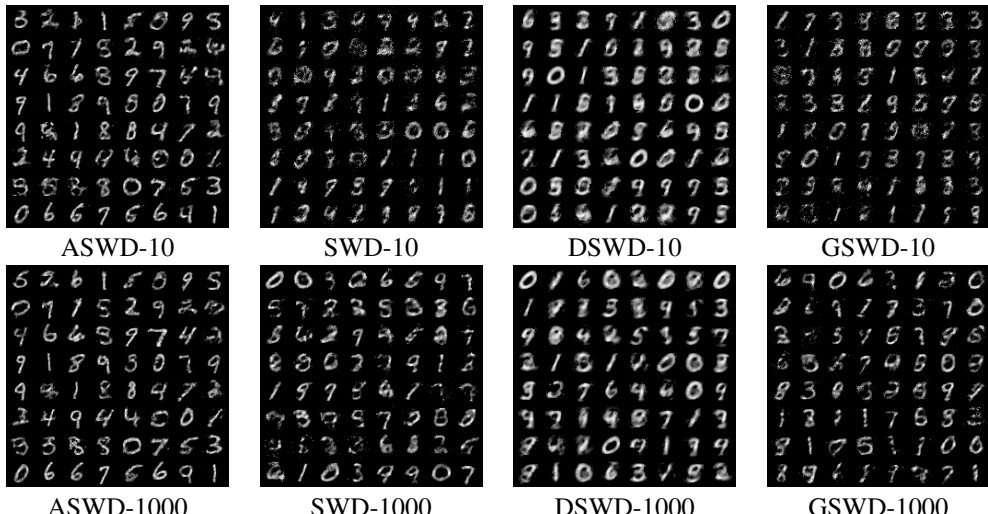

Figure 13: Randomly selected samples generated by different metrics, 10 and 1000 refer to the number of projections.

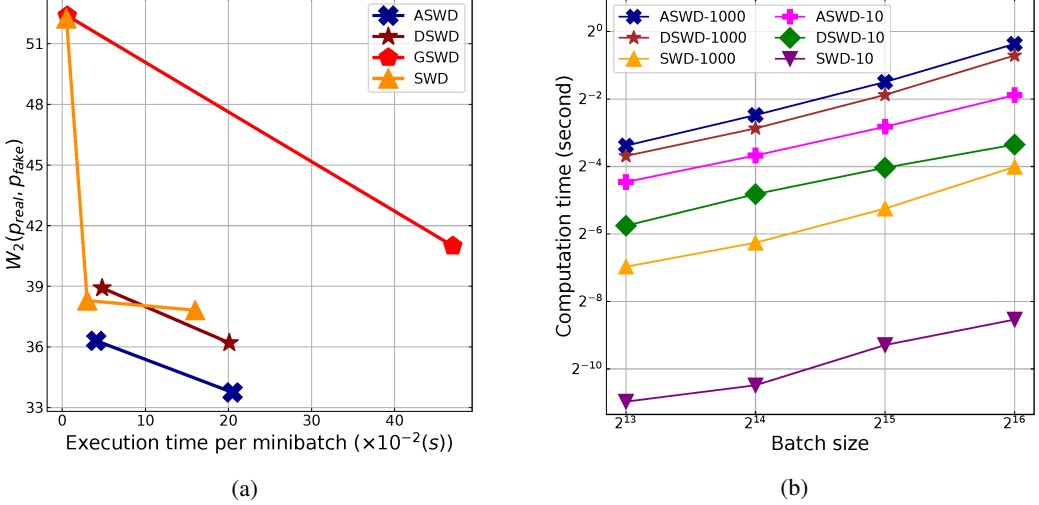

(a)

(b)

Figure 14: (a) The execution time of different methods and the 2-Wasserstein distance between the real images and the fake images generated by their corresponding models. Each dot of the curve of SWD corresponds to the performance of the SWD with the number of projections $L = \{10, 1000, 10000\}$, in sequence. Each dot of the other curves correspond to the performance of the other methods with the number of projections $L = \{10, 1000\}$, in sequence. (b) Computation cost of calculating different metrics, the horizontal axis is the number of samples from the compared distributions, and the horizontal axis is the averaged time consumption for calculating the distances. It can be found that the evaluated metrics tend to scale in $\mathcal{O}(N \log N)$.

APPENDIX I    SLICED WASSERSTEIN AUTOENCODERS

We train an autoencoder using the framework proposed in (Kolouri et al., 2019b), where an encoder and a decoder are jointly trained by minimizing the following objective:

$$\min_{\phi,\psi} \mathrm{BCE}(\psi(\phi(x)),x) + L_1(\psi(\phi(x)),x) + \mathrm{SWD}(p_z,\phi(x)), \tag{55}$$

where $\phi$ is the encoder, $\psi$ is the decoder, $p_z$ is the prior distribution of latent variable, $\mathrm{BCE}(\cdot,\cdot)$ is the binary cross entropy loss between reconstructed images and real images, and $L_1(\cdot,\cdot)$ is the L1 loss between reconstructed images and real images. We train this model using different slice-based Wasserstein metrics, including the ASWD, the DSWD, the SWD, and the GSWD. Here we use the ring distribution as the prior distribution as shown in Figure 16. We report the binary cross entropy loss during test time and the 2-Wasserstein distance between prior and the encoded latent variable $\phi(x)$ in Figure 15. The slice-based Wasserstein metrics used as the third term in Equation (55) is also recorded at each iteration and presented in Figure 15 in order to analyze the factors that causes the differences in the performance of models trained with different slice-based Wasserstein metrics.

It can be observed from the first two columns of Figure 15 that while the model trained with the ASWD, SWD, and DSWD lead to similar 2-Wasserstein distance between the encoded latent variable distribution and the prior distribution, which implies that they have similar coverage of the prior distribution as shown in Figure 16, the model trained with the ASWD converges slightly faster to smaller binary cross entropy loss than the others. As shown in Figure 15(b) and 15(c), since the obtained GSWDs are trivially small compared with the other metrics, models trained with GSWDs only focuses on the reconstruction loss and therefore produce reconstructed images of higher qualities in terms of the binary cross entropy. However, although models trained with the GSWD polynomial and the GSWD circular lead to smaller binary cross entropy loss than the ASWD, their latent distributions present very different data structures from the specified prior distribution, which can be problematic in certain applications where the support of the latent distribution is required to be within a particular range.

Some MNIST images randomly generated by SWAEs trained with different metrics are given in Figure 17.

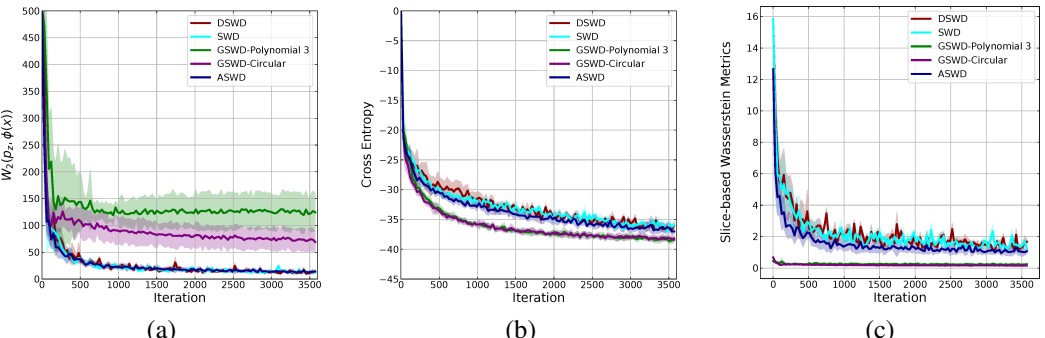

Figure 15: Convergence behavior of SWAEs trained with different slice-based Wasserstein metrics. (a) The 2-Wasserstein distance between the prior distribution $p_z$ and the distribution of encoded feature $\phi(x)$. (b) The binary cross entropy loss between the reconstruction and real data. (c) The slice-based Wasserstein metric used to train the model.

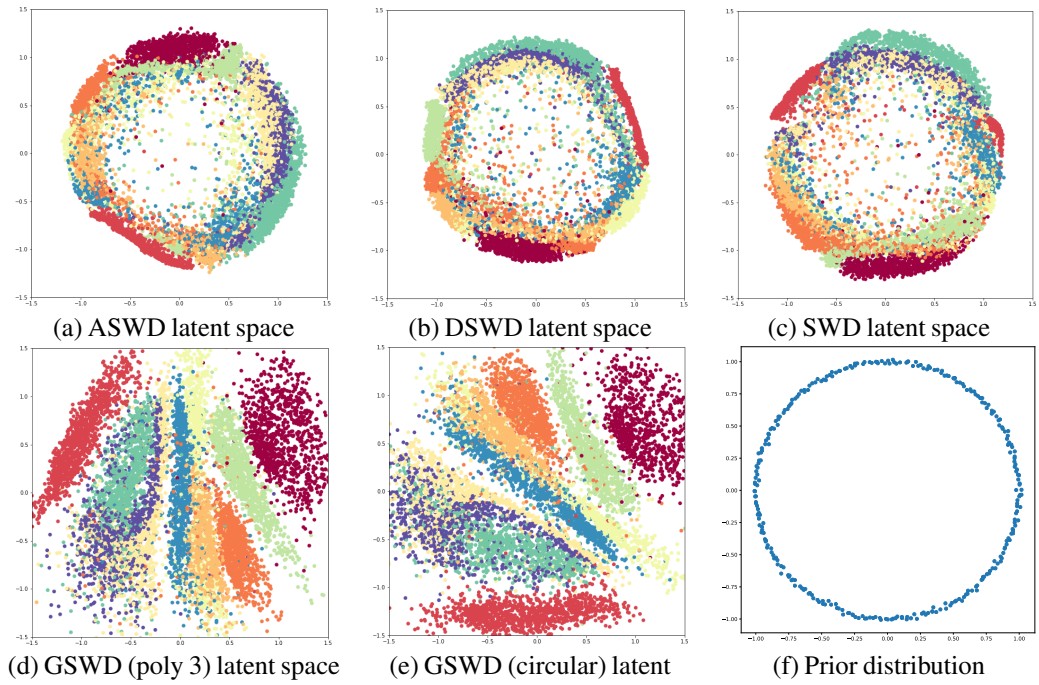

Figure 16: Comparisons between the encoded latent space generated by different slice-based Wasserstein metrics.

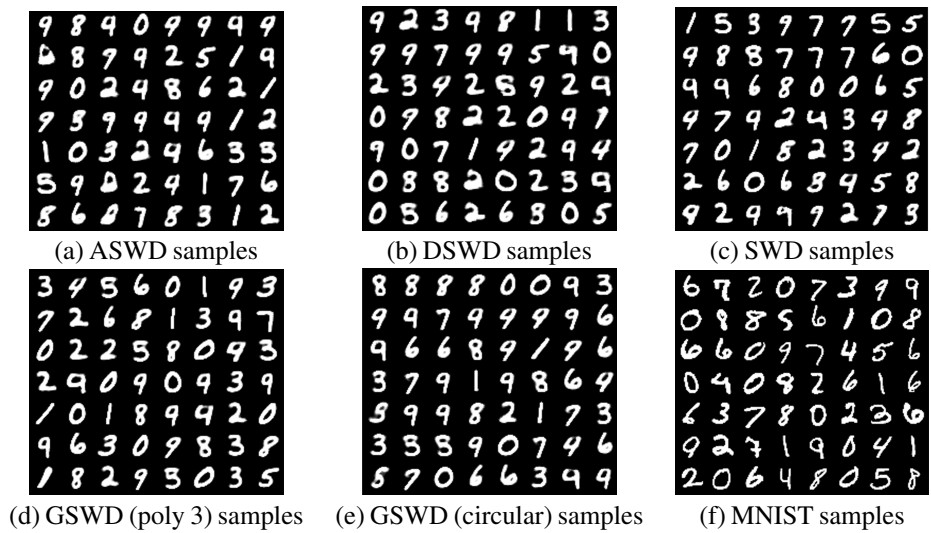

Figure 17: MNIST images randomly generated by SWAEs trained with different metrics.

## APPENDIX J COLOR TRANSFERRING

Color transferring can be formulated as an optimal transport problem (Bonneel et al., 2015; Radon, 1917). In this task, the color palette of a source image is transferred to that of a target image, while keeping the content of source image unchanged. To achieve this, the optimal transport can be used to find the alignment between image pixels by calculating the optimal mapping of color palettes. In this experiment, instead of solving the optimal mapping in the original space, we first project the distribution onto one-dimensional spaces and average the alignment between one-dimensional samples as an approximation of the optimal mapping in the original space. After obtaining the approximation, we replace pixels of the source image with the averaged corresponding pixels in the target image. To reduce the computational cost, we utilize the approach proposed in (Muzellec and Cuturi, 2019), where the K-means algorithm is used to cluster the pixels of both source and target images, and then we implement color transfer for the quantized images whose pixels are consist of the centers of 3000 clusters rather than the original source and target images.

We present the results of color transferring in Figure 18. It can be observed that the ASWD and the DSWD produce sharper images than the SWD and the GSWD (polynomial), we conjecture that is because the ASWD and the DSWD can generate better alignment of pixels. The GSWD (circular) tends to generate high contrast images, but sometimes produces images with low color diversity as in Figure 18(a).The Max-SWD has the highest contrast among all methods, but this is due to it only uses a single projection to obtain the transport mapping, thus there is no need to average different pixels from the target image. A disadvantage of the Max-SWD is that the transferred images generated by Max-SWD is not smooth enough and do not look realistic. The ASWD can generate smooth and realistic images than the SWD and the Max-SWD, even when the number of projections is as small as 10. Transferred images obtained by transferring colors from the source to target using standard optimal transport maps is also presented in Figure 18 for reference (Ferradans et al., 2014).

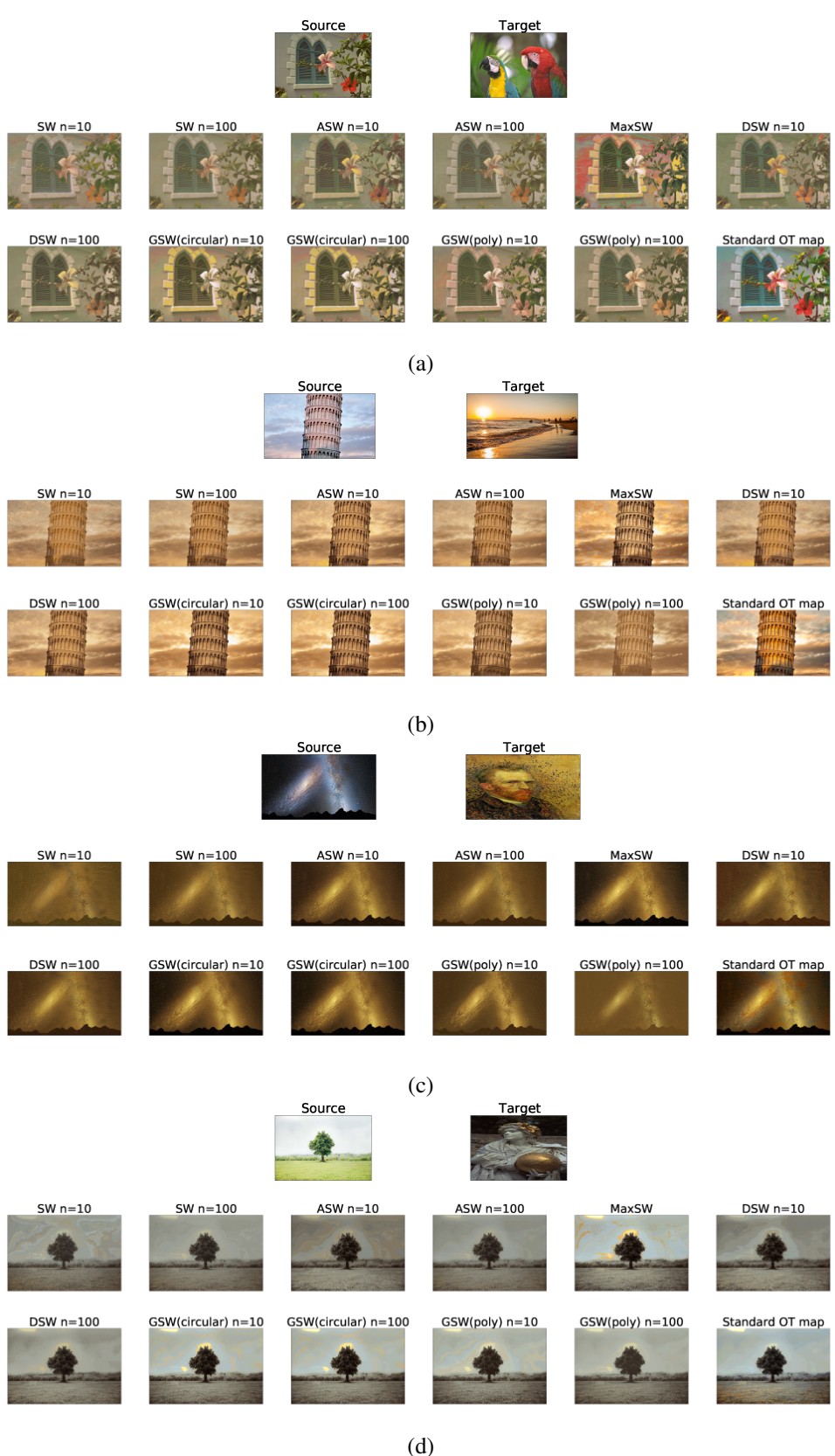

Figure 18: Top rows are source images and target images, lower rows show transferred images obtained by using different methods with different number of projections. Source and target images are from (Bonneel et al., 2015) and https://github.com/chia56028/Color-Transfer-between-Images.

## APPENDIX K    SLICED WASSERSTEIN BARYCENTER

Sliced Wasserstein distances can also be applied in the barycenter calculation and shape interpolation (Bonneel et al., 2015). Here we compare barycenters produced by different slice-based Wasserstein metrics, including the GSWD (circular and polynomial), the ASWD, the SWD, and the DSWD. Specifically, we compute barycenters of different shapes consisting of point clouds, as shown in Figure 19. Each object in Figure 19 corresponds to a specific barycenter with different weights. The Wasserstein barycenters are also presented in Figure 19 for reference, which provide geometrically meaningful barycenters at the expense of significantly higher computational cost .

Formally, a sliced-Wasserstein barycenter of objects $\mu = \{\mu_1, \mu_2, \cdots, \mu_N \in P_k(\mathbb{R}^d)\}$ assigned with weights $w = [w_1, w_2, \cdots, w_N \in \mathbb{R}]$ is defined as:

$$\text{Bar}(\mu, w) = \underset{\mu \in P_k(\mathbb{R}^d)}{\text{argmin}} \sum_{i=1}^{N} w_i \text{SWD}(\mu, \mu_i). \tag{56}$$

In this experiment, we set $N = 3$ and compute barycenters corresponding to different weights. The results are given in Figure 19.

From Figure 19, it can be observed that the ASWD produces similar barycenters as that of the SWD, which are sharper than the DSWD, and more meaningful than the GSWD (polynomial). The flexibility of the injective neural networks $g(\cdot)$ and its optimization in the ASWD can be potentially combined with specific requirements in particular tasks to generate calibrated barycenters - we leave this as a future research direction.

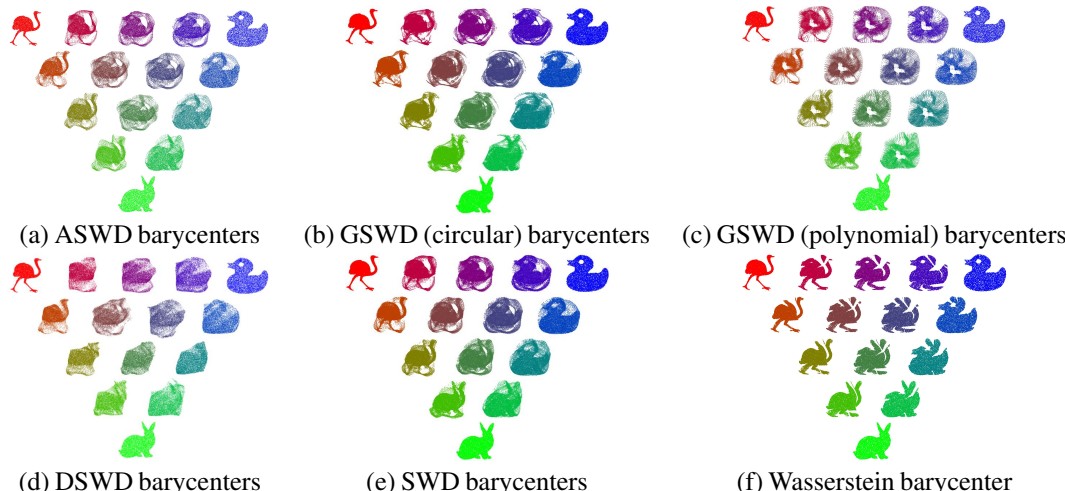

(a) ASWD barycenters    (b) GSWD (circular) barycenters    (c) GSWD (polynomial) barycenters

(d) DSWD barycenters    (e) SWD barycenters    (f) Wasserstein barycenter

Figure 19: Sliced Wasserstein barycenters generated by the ASWD, the GSWD, the DSWD, and the SWD, and the Wasserstein barycenter.

