# OpenReview forum: "Augmented Sliced Wasserstein Distances"
_ICLR.cc/2022/Conference — ICLR 2022 Poster_

### Official Review · Reviewer_6LWm · 2021-11-01

**Correctness:** 4
**Technical Novelty And Significance:** 3
**Empirical Novelty And Significance:** Not applicable
**Recommendation:** 6
**Confidence:** 3

**Main Review:**


This paper is overall well written, and ASWD is well motivated.
The proposed method seems to have some originality, though the technique of using non-linear functions to improve the efficiency of slicing has been proposed in GSWD-NN.
The author theoretically show that their proposed method is a valid distance metric even if they use neural networks for the non-linear functions.
I did not check the theoretical results carefully, but they are correct as far as I can tell.
The significance of ASWD has been shown through various experiments such as flow, generalize model, barycenter, and color transfer.

My concern is about the difference between ASWD and GSWD-NN.
To my understanding, the main differences between them are
(1) GSWD-NN is not a valid metric because they uses non-injective function, and (2) ASWD calculates SWD after the non-linear mappings, while max-GSWD-NN directly calculates 1-dimensional Wasserstein distance after the mapping.

Questions:
- (a) As the authors claim, GSWD-NN is not a valid metric. However, if they use an injective function, would GSWD-NN be a valid metric? In other words, is the claim against the inappropriate choice of $g(x)$? Or is GSWD inherently difficult to combine with NNs?
- (b) The second question is about the cause of the difference in performance. The experiment in Sec G.2 indicates that there is not much difference between ASWD and ASWD-non-injective in performances when optimizing NN. So is the difference in performance between ASWD and GSWD-NN due to the random projection (or some other reasons, e.g., regularization term $L_\lambda$)?


**Summary Of The Paper:**

This paper proposes a variant of sliced Wasserstein distance, named augmented sliced Wasserstein distance.
ASWD maps input data points to hypersurfaces using neural networks, then calculates SWD on the hypersurfaces.
ASWD alleviates the low efficiency problem of SWD for high-dimensional data.
Various tasks including flow, generative modeling, and barycenters show the advantage of AWSD against some existing methods.

**Summary Of The Review:**

This paper is well written and well motivated. The proposed method seems to have some originality, though the technique of using non-linear functions has been proposed. Various experiments show its significance over existing methods.

---

> ### Author Response · Authors · 2021-11-20
> **Response to Reviewer 6LWm (1/2)**
>
> Many thanks for your positive and constructive feedback! Please see below our response to your comments:
>
> **(1) As the authors claim, GSWD-NN is not a valid metric. However, if they use an injective function, would GSWD-NN be a valid metric? In other words, is the claim against the inappropriate choice of $g(x)$? Or is GSWD inherently difficult to combine with NNs?**
>
> **[As the authors claim, GSWD-NN is not a valid metric. However, if they use an injective function, would GSWD-NN be a valid metric?]** Even if injective functions are used in the GSWD-NN, it is still not a valid metric. The reason is that GSWD-NN breaks the framework of the SWD, by directly using the outputs of neural networks as the projections of samples. And due to this construction, we can find examples showing that the GSWD-NN with injective mappings is not a valid metric. For instance, let $g$ be the identity function $g(x)=x$, which is clearly an injection, and $\mu$, $\nu\in P_k(\mathbb{R}^2)$ be two distinctive Dirac measures $\mu(\mathcal{X})=\frac{1}{2}\sum_{n=1}^2\int_\{\mathcal{X}\}\delta(x-x_n)dx$ and $\nu(\mathcal{Y})=\frac{1}{2}\sum_\{n=1\}^2\int_\{\mathcal{Y}\}\delta(y-y_n)dy$ for $\mathcal{X},\mathcal{Y}\subseteq\mathbb{R}^2$, where
>
> $
> \begin{align}
> x_1 &= \begin{bmatrix}
>        -1 \\\\
>        -2
>      \end{bmatrix}, x_2 = \begin{bmatrix}
>        2 \\\\
>        1
>      \end{bmatrix},(1)\\\\
>      y_1&= \begin{bmatrix}
>        2 \\\\
>        -2
>      \end{bmatrix}, y_2 = \begin{bmatrix}
>        -1 \\\\
>        1
>      \end{bmatrix}. (2)
> \end{align}
> $
>
> To compute the GSWD-NN, the samples $x_1$, $x_2$ and $y_1$, $y_2$ need to be sorted in an ascending order. We use $x'_n$ and $y'_n$ to denote the sorted samples. The samples after sorting become:
>
> $
> \begin{align}
> x'_1 &= \begin{bmatrix}
>        -1 \\\\
>        -2
>      \end{bmatrix}, x'_2 = \begin{bmatrix}
>        2 \\\\
>        1
>      \end{bmatrix},(3)\\\\
>      y'_1 &= \begin{bmatrix}
>        -1 \\\\
>        -2
>      \end{bmatrix}, y'_2 = \begin{bmatrix}
>        2 \\\\
>        1
>      \end{bmatrix}. (4)
> \end{align}
> $
>
> Then the GSWD-NN equals the average distance between the sorted samples given in Equation (3) and (4). As illustrated above, while $\mu\neq \nu$, GSWD-NN$(\mu, \nu)=0$, which implies the GSWD-NN defined with the injection $g(x)=x$ does not satisfy the identity of indiscernibles and thus is not a valid metric.
>
> **[In other words, is the claim against the inappropriate choice of $g(x)$?]** As we discussed in the response above, even with an injective mapping $g(\cdot)$, the resulting distance of the GSWD-NN is still not a valid metric. Hence, the claim is not against the inappropriate choice of $g(x)$ in GSWD-NN.
>
> **[Or is GSWD inherently difficult to combine with NNs?]** Yes, as stated in  (Kolouri et al., 2019), it is highly non-trivial to show the injectivity of the associated generalized Radon transform and consequently the metricity of the GSWD-NNs. On one hand, as in our response to the Reviewer ViRm, there are some non-trivial conditions for designing the defining functions of generalized Radon transforms, so it is highly non-trivial to incorporate neural networks into the defining functions of generalized Radon transforms and the corresponding GSWDs. On the other hand, (Nguyen et al., 2021)  proposed an alternative approach to combine GSWDs with neural networks. Specifically, (Nguyen et al., 2021)  proposed a clever design of using neural networks to search for the best distribution of important directions and thus improve the projection efficiency of GSWDs. Yet, the resulting DGSWD is still not able to produce nonlinear projections adaptively as one need to first specify a suitable generalized Radon transform; In comparison, ASWD learns nonlinear projections adaptively that can capture the complex structure of the data distributions.
>
> **(2) The second question is about the cause of the difference in performance. The experiment in Sec G.2 indicates that there is not much difference between ASWD and ASWD-non-injective in performances when optimizing NN. So is the difference in performance between ASWD and GSWD-NN due to the random projection (or some other reasons, e.g., regularization term $L_\lambda$)?**
>
>
> **[The experiment in Sec G.2 indicates that there is not much difference between ASWD and ASWD-non-injective in performances when optimizing NN.]** Although the ASWD-non-injective exhibits similar performance as the ASWD, we observed in the ablation study in Appendix G.2 of the paper that ASWD-non-injective can lead to more unstable training which is evident in Figure 5.

---

> > ### Author Response · Authors · 2021-11-20
> > **Response to Reviewer 6LWm (2/2)**
> >
> > **[Is the difference in performance between ASWD and GSWD-NN due to the random projection (or some other reasons, e.g., regularization term $L_\lambda$)?]**
> > As you sharply pointed out, GSWD-NN directly uses the outputs of neural networks as the projections of sample which are not considered random projections (more details in the following paragraph) and is not a valid metric; while ASWD is within the framework of SWDs which generate random projections and is a valid metric. One main advantage of the ASWD is that it learns data
> > patterns and produce well-separated projections with random projection directions, as one can observe in Figures 1(a) and 1(c) in the paper.
> > The difference in performance between ASWD and GSWD-NN can hence be due to several factors, including the high projection efficiency of the ASWD resulted from being data-adaptive and its favorable theoretical properties as shown in Theorem 1 and Corollary 1.1.
> >
> > The framework of GSWD-NNs directly uses the output of the neural network as the slicing/projection results. It has two implications: 1) There is only a fixed number of projections used in the GSWD-NN, which is equal to the number of nodes in the neural
> > network’s output layer; 2) there is no random projection involved, unlike SWD, GSWD and ASWD. The first implication restricts the flexibility of the GSWD-NN, as one needs to re-train a neural network when changing the number of projections $L$. The second point breaks the framework of the slice-based Wasserstein metrics, which makes the GSWD-NN loses its metric properties and hard to interpret. To conclude, compared with the GSWD-NN, the advantage of the ASWD is that it inherits the favourable properties of slice-based Wasserstein distances while being data-adaptive, hence the ASWD presents better performance than the GSWD-NN. Notably, we want to emphasize that the randomness introduced by random projections does not lead to random experiment results of ASWD, as the ASWD consistently produced superior performance in multiple Monte Carlo runs of the experiments included in the paper.
> >
> > We want to clarify that the regularization term (Equation (19)) does not apply to other slice-based Wasserstein metrics. Specifically, the regularization term is applied to constrain the parameterized mapping
> > $g_\omega(\cdot)$, while this mapping is not used in other methods except GSWD-NN. To support our argument, in the revised paper, the effect of the regularization term on the performance of Max-GSWD-NN was investigated in Appendix G.2, where the performance of the Max-GSWD-NN and the Max-GSWD-NN optimized with the regularization term used in the ASWD are compared. Specifically, we added in the subsection **"Impact of the regularization coefficient"** of Appendix G.2 the following contents:
> >
> > "In addition, the effect of the regularization term on the performance of Max-GSWD-NN was also investigated in this ablation study. The performance of the Max-GSWD-NN and the Max-GSWD-NN trained with the regularization term used in the ASWD are compared in Figure 8. From the numerical results presented in Figure~8, the Max-GSWD-NN 1 with regularization leads to performance similar to the Max-GSWD-NN 1 without regularization, implying that the performance gap between the ASWD and the Max-GSWD-NN is not due to the introduction of the regularization term.".
> >
> > **References**
> >
> > S. Kolouri, K. Nadjahi, U. Simsekli, R. Badeau, and G. Rohde. Generalized sliced Wasserstein distances. In *Proc. Advances in Neural Information Processing Systems (NeurIPS)*, pp. 261–272, Vancouver,Canada, 2019.
> >
> > K. Nguyen, N. Ho, T. Pham, and H. Bui. Distributional sliced-Wasserstein and applications to generative modeling. In *Proc. International Conference on Learning Representations (ICLR)*, Vienna, Austria, 2021.

---

### Official Review · Reviewer_1Q3z · 2021-11-02

**Correctness:** 4
**Technical Novelty And Significance:** 3
**Empirical Novelty And Significance:** 3
**Recommendation:** 6
**Confidence:** 3

**Main Review:**

- Overall, the paper is well written. It thoroughly  reviews the background and main works on SWD.  The rationale behind the proposed method is justified and aims to design an appropriate non-linear projection mapping that renders the induced sliced-Wasserstein distance (SWD) a valid metric. Theoretical analyses of conditions under which the metric property is preserved are provided.  In this regard the contribution is of interest and significative. Intensive empirical evaluations support these developments and show how the approach contributes efficient SWD computation.

- Specifically,  instead of seeking a generalized Radon Transform (defined by non-linear defining functions that have to follow stringent constraints) or finding the distribution of (non-linear) random projections that maximize the expected 1D Wasserstein distances, the main idea is rather to learn a non-linear mapping function $g$ onto a high-dimension hyper-surfaces. The function $g$ acts as a measurable mapping that generates push-forward measures of the distributions to be compared. Hence, linear random projections onto 1D are performed  to obtain the related augmented SWD (ASWD) as expected 1D Wasserstein distances. The procedure results in defining a spatial Radon transform based on $g$ to compute the SWD which is clearly a clever idea. Therefore, establishing that ASWD is a metric simply derives from the injective property $g$. This facilitates the theoretical study of ASWD.

- As an another interesting contribution, learning $g$ from data is achieved by maximizing a regularized version of ASWD. The range of the regularization parameter $\lambda$ ensuring that the obtained $g$ leads to a valid metric  is stated. To guarantee the injection property, a neural network (NN) is considered and $g$ is get as the concatenation of the input and the output of the NN. The idea inspired from DenseNet is tricky and presents a clear advantage of avoiding learning complex NN.

- Intensive empirical evaluations are conducted. They show impressive results compared to existing methods as SWD, distributional SWD (DSWD), generalized SWD (GSWD). In a synthetic problem of evolving source distribution to target one, ASWD achieves the smallest distance. The ASWD is also used to train a generative model for image generation. The reported performances (FID score) highlight the effectiveness of ASWD.

- ASWD is shown to be a valid metric. What are the topological properties of ASWD compared to plain Wasserstein distance, SWD? Beyond empirical evaluations, can one theoretically relate ASWD to WD and SWD?

- The main feature of ASWD is to project the samples onto a $d_\theta$-high-dimensional hyper-surface using $g$.  Due to the design of $g$ in Eq. 19, $d_\theta > d$. Does the choice of $d_\theta$ impact the quality of the obtained ASWD?

- How evolve the computation time according to the batch size? In Nguyen et al., 2021 the computation complexity tends to scale in $O(n \lg n)$.

- Unless mistaken, it seems that the FID scores in Table 1 for CIFAR 10 and CelebA are not competitive with the ones reported on Figure 3 in Nguyen et al, 2021. Indeed Nguyen et al. 2021 show smaller FID of order 60 for CIFAR 10 and 70 for CelebA. Can the authors explain the shift in the reported performances in Table 1?


Minor comments
- In remark 4, the expression 'rank of the augmented' is not clear and is undefined.

*After rebuttal*

I read authors rebuttal. They address most  raised points. Indeed they study the influence of $d_\theta$ on  the computed ASWD as long as the computational complexity wich is similar to the one  of distributional SWD up to a constant. Also the mismatch in the reported FID scores is clarified and  fixed. One left out concern is  the topological property of ASWD.

**Summary Of The Paper:**

The paper proposes a new slice-based approach to efficiently compute the Wasserstein distance between two distributions $\nu$ and $\mu$. The method termed ASWD (augmented sliced Wasserstein Distance) first  projects the samples from $\nu$ and $\mu$ onto a higher dimensional space using a non linear injective mapping function and then uses the classical random linear projections onto 1D to compute the sliced Wasserstein Distance. Overall, the procedure amounts to applied a spatial Radon Transform to perform the slicing. Theoretical results establish conditions  under which ASWD is a metric. A numerical algorithm is given along with the design of the injective mapping using NN. Empirical evaluations on simulation datasets and on generative modeling highlight the potential of the proposed method over existing approaches.

**Summary Of The Review:**

The paper is well written and proposes a new sliced Wasserstein distance using the spatial Radon defined Transform defined as line integrals of a function along all hyperplances over a hyper-surface induced by a non-linear injective mapping. The new distance is supported by theoretical analysis and empirical experiments showing its effectiveness. Topological properties of the distance and some details on the empirical  evaluations remain to be clarified.

---

> ### Author Response · Authors · 2021-11-20
> **Response to Reviewer 1Q3z (1/2)**
>
> Many thanks for your constructive and insightful comments! Please see below our detailed response to your comments and we hope that it will resolve any concern you had:
>
> **(1) ASWD is shown to be a valid metric. What are the topological properties of ASWD compared to plain Wasserstein distance, SWD? Beyond empirical evaluations, can one theoretically relate ASWD to WD and SWD?**
>
> The theoretical connection between the SWD and the ASWD can be found in Remark 2. In particular, the ASWD between the compared measures $\mu$ and $\nu$ can be interpreted as the SWD between the transformed push-forward measures $\hat{\mu}=g_\\#\mu$ and $\hat{\nu}=g_\\#\nu$.
>
> $
> \begin{aligned}
> \operatorname{ASWD}_\{k\}(\mu, \nu ; g)&=\left(\int_\{\mathbb{S}^{d_\{\theta}-1}\}W_\{k\}^{k} \left(\mathcal{R}_\{\hat{\mu}_\{g\}\}(\cdot, \theta), \mathcal{R}_\{\hat{\nu}_\{g\}\}(\cdot, \theta)\right) d \theta\right)^{\frac{1}{k}} (1)\\\\
> &=\operatorname{SWD}_\{k\}\left(\hat{\mu}_\{g\}, \hat{\nu}_\{g\}\right).
> \end{aligned}
> $
>
> It is non-trivial to derive topological properties of variants of SWDs with non-linear projections such as ASWD with arbitrary injective neural networks. We thank the reviewer for the comment and will consider this as an important future research direction, which we have now specified in the conclusion of the revised paper draft.
>
> **(2) The main feature of ASWD is to project the samples onto a $d_\theta$-high-dimensional hyper-surface using $g$. Due to the design of $g$ in Eq. 19, $d_\theta>d$. Does the choice of $d_\theta$ impact the quality of the obtained ASWD?**
>
> We have now investigated the impact of the dimensionality $d_\theta$ of the augmented space on the performance of the ASWDs and have reported the numerical results in Appendix G.2 of the revised paper.
>
> Specifically, the subsection of **"Choice of the dimensionality $d_\theta$ of the augmented space"** is added to Appendix G.2, which reports that "To investigate how the dimensionality $d_\theta$ of the augmented space affects the performance of the ASWD, different choices of $d_\theta$ are employed in the ASWD. Specifically, the injective network architecture $g_\omega(x)=[x, \phi_\omega(x)]: \mathbb{R}^d\rightarrow\mathbb{R}^{d_\theta}$ given in Equation (18) is adopted and $\phi_\omega$ is set to be single fully-connected neural networks whose output dimension equals \{1, 2, 3, 4\} times its input dimension, i.e. $d_\theta=\\{2d, 3d, 4d, 5d\\}$, respectively, where $d$ is the dimensionality of $x$. The numerical results are presented in Figure 10, and it can be found that the ASWDs present similar results across different choices of $d_\theta$. It can also be observed in Figure 10 that the ASWDs with different choices of $d_\theta$ consistently produce better performance than the other evaluated slice-based Wasserstein metrics.''
>
> **(3) How evolve the computation time according to the batch size? In (Nguyen et al., 2021), the computation complexity tends to scale in $\mathcal{O}(N\log N)$.**
>
> We have compared the effect of batch size on the computation cost of the ASWD, the DSWD, and the SWD. Specifically, in Appendix H.2 of the revised paper, the computation time of the evaluated methods with $L=\\{10, 1000\\}$ projections and $N=\\{2^{13}, 2^{14}, 2^{15}, 2^{16}\\}$ samples are reported in Figure 14b. From the results presented in Figure 14 (b), we observed that, similar to the computational complexity of the DSWD and the SWD, the computational complexity of the ASWD empirically tends to scale in $\mathcal{O}(N\log N)$.

---

> > ### Author Response · Authors · 2021-11-20
> > **Response to Reviewer 1Q3z (2/2)**
> >
> > **(4) Unless mistaken, it seems that the FID scores in Table 1 for CIFAR 10 and CelebA are not competitive with the ones reported on Figure 3 in (Nguyen et al., 2021). Indeed (Nguyen et al., 2021) show smaller FID of order 60 for CIFAR 10 and 70 for CelebA. Can the authors explain the shift in the reported performances in Table 1?**
> >
> > Thank you very much for pointing out the inconsistency here. In fact, the generative modelling experiment results we presented in the paper were based on the implementation associated with the first arXiv version of (Nguyen et al., 2021) (**https://arxiv.org/abs/2002.07367v1**) which led to higher FID scores. And indeed in the latest, published version (**https://openreview.net/pdf?id=QYjO70ACDK**), the reported FID scores are significantly smaller.
> > The numerical results reported in the earlier version and the latest version of (Nguyen et al., 2021) are shown in Table 1 below for reference. And we identified that the differences were caused by that the latent variable's dimension was changed from 32 to 100, and data normalization was used as a pre-processing technique in the dataloader in the latest version.
> >
> > We have updated our implementation according to the latest repository of (Nguyen et al., 2021): **https://github.com/VinAIResearch/DSW** and updated the experiment results reported in Table 1, Figure 3, and Figure 11 in the revised paper draft. To compare with the results reported in (Nguyen et al., 2021) as shown in Table 1 below in the response, we also list the updated results (as shown in Table 1 in the updated paper draft) in Table 2 below. From Table 2, it can be observed that all claims in Section 5.2 still hold, i.e. the ASWD leads to significantly improved FID scores among all evaluated distances metrics on both datasets, with the same number of projections, which implies that images generated with the ASWD are of higher qualities in this experiment.
> >
> > |  |  |  |  |  |  |  |
> > | :---: | :---: | :---: | :---: | :---: | :---: | :---: |
> > | **Earlier version (https://arxiv.org/abs/2002.07367v1)** |
> > |  | | CIFAR |  |  | CELEBA |  |
> > | L | SWD | GSWD | DSWD | SWD | GSWD | DSWD |
> > | 100 | 150.64$\pm$7.64 | N/A | 115.17$\pm$13.2 | 96.43$\pm$2.74 | N/A | 90.74$\pm$3.03 |
> > | 1000 | 144.75$\pm$4.33 | N/A | 112.81$\pm$4.87 | 93.17$\pm$2.14 | N/A | 88.25$\pm$3.56 |
> > | 10000 | 120.03$\pm$3.38 | N/A | 100.28$\pm$5.67 | 91.22$\pm$3.35 | N/A | 88.13$\pm$1.27 |
> > |**Latest version (https://openreview.net/pdf?id=QYjO70ACDK)** |
> > |  | | CIFAR |  |  | CELEBA |  |
> > | L | SWD | GSWD | DSWD | SWD | GSWD | DSWD |
> > | 100 | 109.7$\pm$5.64 | 103.11$\pm$6.92 | 62.83$\pm$6.24 | 90.11$\pm$10.11 | 87.18$\pm$8.97 | 75.94$\pm$5.54 |
> > | 10000 | 98.61$\pm$3.62 | 93.51$\pm$6.12 | 56.42$\pm$3.78 | 82.02$\pm$6.33 | 84.22$\pm$7.93 | 66.85$\pm$7.22 |
> >
> > **Table 1: The reported FID scores in the earlier and the latest version of (Nguyen et al., 2021).**
> >
> > ||||||
> > | :---: | :---: | :---: | :---: | :---: |
> > |||**CIFAR**|||
> > | L | SWD | GSWD | DSWD | ASWD |
> > | 10 | 121.394$\pm$6.99 | 108.32$\pm$5.58 | 74.17$\pm$3.12 | 65.66$\pm$3.16 |
> > | 100 | 104.61$\pm$5.20 | 105.21$\pm$3.21 | 66.53$\pm$3.87 | 62.51$\pm$1.89 |
> > | 1000 | 102.33$\pm$5.34 | 98.18$\pm$5.06 | 62.28$\pm$5.67 | 59.33$\pm$3.22 |
> > |||**CELEBA**|||
> > | L | SWD | GSWD | DSWD | ASWD |
> > | 10 | 94.81$\pm$2.48 | 95.12$\pm$4.21 | 86.03$\pm$1.43 | 81.22$\pm$1.34 |
> > | 100 | 88.73$\pm$5.66 | 86.71$\pm$3.54 | 76.13$\pm$3.45 | 73.15$\pm$2.56 |
> > | 1000 | 86.54$\pm$4.10 | 85.22$\pm$6.26 | 71.29$\pm$4.70 | 67.37$\pm$2.11 |
> >
> > **Table 2: FID scores after updating the implementation to be consistent with the latest version of (Nguyenet al., 2021).**
> >
> > **(5) In remark 4, the expression "rank of the augmented" is not clear and is undefined.**
> >
> > The rank of the augmented space refers to the intrinsic dimension of the augmented space, i.e. the number of non-zero dimensions in $\phi_\omega(\cdot)$. To clarify this, we have revised the last sentence in Remark 4 from:
> >
> > "The rank of the augmented space is hence explicitly controlled
> > by the flexible choice of $\phi_\omega(\cdot)$ and implicitly regularized by $L(\mu,\nu,\lambda; g_\omega)$."
> >
> > to:
> >
> > "The intrinsic dimension of the augmented space, i.e. the number of non-zero dimensions in the augmented space, is hence explicitly controlled
> > by the flexible choice of $\phi_\omega(\cdot)$ and implicitly regularized by $L(\mu,\nu,\lambda; g_\omega)$.".
> >
> > **References**
> >
> > K. Nguyen, N. Ho, T. Pham, and H. Bui. Distributional sliced-Wasserstein and applications to generative modeling. In *Proc. International Conference on Learning Representations (ICLR)*, Vienna, Austria, 2021.

---

> > > ### Comment · Reviewer_1Q3z · 2021-11-27
> > > **On the rebuttal 2/2**
> > >
> > > Thanks  for clarifying the inconsistency of the FID scores of DSWD. For CelebA the results in Table 2 above are consistent with  the ones of (Nguyen et al., 2021). Regarding CIFAR,  their reported results are rather competitive compare to ASWD (and contrary to the FID in Table 2). Probably, this might be due  to implementation details or random seed selection.

---

> > ### Comment · Reviewer_1Q3z · 2021-11-27
> > **On the rebuttal**
> >
> > Thanks to the authors for providing more detailed insights in the proposed approach. Though ASWD induces a slight increase in computation time as to DSWD, it scales similarly in $\mathcal{O}(N logN)$. Apparently, ASWD  does not heavily hinge  on  the choice of $d_\theta$. According to the expertise of the authors, can one transfer this into a rule of thumb for the  user to  set $d_\theta$?

---

> > > ### Author Response · Authors · 2021-11-28
> > > **Choice of the dimensionality of the augmented space**
> > >
> > > Yes, based on the numerical results obtained, the performance of the ASWD is found to be not sensitive to the choice of $d_\theta$. From the results presented in Figure 10, a rule of thumb for the user to set $d_\theta$ is to set $d_\theta=2d$. This choice of $d_\theta$ produces good performance of the ASWD and is computationally cheaper than all other values of $d_\theta$ we evaluated.

---

### Official Review · Reviewer_RRsR · 2021-11-03

**Correctness:** 4
**Technical Novelty And Significance:** 3
**Empirical Novelty And Significance:** 3
**Recommendation:** 6
**Confidence:** 4

**Main Review:**

#### Strength:
The paper is well-written and the approach is mostly well presented. The authors introduce the ASWD and derive a set of theoretical properties and propose a numerical algorithm to approximate ASWD. They finally conduct several experiments to illustrate the advantages of ASWD over the state-of-the-art of sliced-based Wasserstein distance like SWD, GSWD, and DSWD.
The code is given, results seem reproducible.

#### Weakness:

- The authors argue that projecting the original data samples of the source and target distributions into higher-dimensional space leads to maximally discriminating projected samples, by constraining the $k$-th moments of the projected samples. In my opinion, the latter constraint could be well motivated and clarified (or at least well explained or discussed).

- Since the projection is made by the same mapping $g$, It might be not guaranteed maximal discrimination over the projected samples. Intuitively, I think the mapping $g$ should be data-dependent and/or making a trade-off with different regularization parameters, i.e. the functional $L(\mu, \nu, \lambda; g)$ becomes $L(\mu, \nu, \lambda_1, \lambda_2; g)$.

- As was shown in Remark 3 that if the tuning parameter $\lambda>1$ then ASWD is a proper metric, whereas the authors conducted their experiments with values $\lambda<1$, because it leads to outperforming the other sliced-based Wasserstein metrics. In my opinion, this restricts the applications of ASWD, since one has to investigate a panel of tuning parameters $\lambda<1$ to first check if ASWD is still a valid distance then if it gives good performance compared to the other SWDs.

- The non-linearity of the injective mapping $g(\cdot) = [Id, \phi_\omega(\cdot)]$ seems a simple injective NN, allowing to gain efficient time complexity. Does it possible to construct $g$ by other injective NNs? It is probably that the fact for a small $lambda <1$ ASWD outperforms the sliced-based Wassrstein distance is linked to the choice of this special parameterization of the injective NN.

- How is the dimensionality of the latent space $d_\theta$ decided and is its impact? The paper does not seem to have included discussions on this seemingly important hyperparameter.

#### Typo:
- page 4: (Definition 1) "Given an measurable" -> "Given a measurable"

**Summary Of The Paper:**

This paper introduces the augmented sliced Wasserstein distance (ASWD), a new variety of sliced Wasserstein distance (SWD), that allows comparing two probability distributions by combining a nonlinear embedding of the sample data points to a higher-dimensional space with a slicing scheme to calculate 1D Wasserstein distances between uniformly projection directions. The authors introduce the spatial Radon transform, which includes the standard Radon transform and the special case of polynomial generalized Random transform (introduced in Kolouri et al. 2019). They further prove that ASWD is a valid metric if and only if the mapping is injective. Several experiments are conducted on generative modelling (CIFAR10, CelebA,  MNIST, color transferring).

**Summary Of The Review:**

The authors introduce the ASWD a new method of SWDs and derive a set of theoretical properties and propose a numerical algorithm to approximate ASWD. The ASWD is a proper metric if and only if the original data samples embedding in a higher-dimensional space through a nonlinear injective mapping. In my opinion, the optimizing ASWD with respect to the injective mapping is strongly depending on the choice of the tuning parameter which guarantees that ASWD is a valid metric if it is greater than $1$. However, in practice, it is better to conduct experiments with values less than $1$. This creates some "broken pieces" between theoretical fundings and experiments.

---

> ### Author Response · Authors · 2021-11-20
> **Response to Reviewer RRsR (1/3)**
>
> Many thanks for your constructive and insightful feedback! Please see below for our detailed response to your comments which we hope would resolve any concerns you had:
>
> **(1) The authors argue that projecting the original data samples of the source and target distributions into higher-dimensional space leads to maximally discriminating projected samples, by constraining the $k$-th moments of the projected samples. In my opinion, the latter constraint could be well motivated and clarified (or at least well explained or discussed).**
>
> **[The authors argue that projecting the original data samples of the source and target distributions into higher-dimensional space leads to maximally discriminating projected samples, by constraining the $k$-th moments of the projected samples.]** We would like to clarify that the hypersurfaces where the source and target distributions are projected onto are not optimized by solely constraining the $k$-th moments of the transformed samples. Instead, as mentioned in **"Optimization objective"** of Section 3.2, the hypersurfaces are optimized by maximizing the objective given by Equation (19), which consists of the distance term and the $k$-th moments term. Again in Section 3.2, we want to clarify that the dissimilar projected samples are obtained by maximizing the first term on the R.H.S of Equation (19), which is the distance between the projected samples, while the constraint on the $k$-th moments of the output of $g(\cdot)$ is a regularization term used to prevent the projections from being arbitrarily large.
>
> **[In my opinion, the latter constraint could be well motivated and clarified.]** Thanks for your insightful comments. As discussed in the last sentence of **"Optimization objective"** in Section 3.2, the constraint on the $k$-th moments of the transformed samples is used to prevent the projections from being arbitrarily large. This particular choice of the regularization term facilitates the proofs to Lemma 2 and subsequently to Corollary 1.1 with details in Appendix D. Besides, we have also examined other types of regularization terms such as the $L_2$ norm of the output of $g(\cdot)$, and empirically they produce similar numerical results as the current regularization term.
>
> We have added to following text after the pseudocode provided in Algorithm 1 (Appendix E) to explain this particular choice of the regularization term:
>
> "In Algorithm 1, Equation~(17) is used as the optimization objective, where the regularization term $L(\mu,\nu,\lambda; g_\omega)=\lambda(\mathbb{E}^{\frac{1}{k}}_\{x\sim \mu\}\big[||g_\omega(x)||^k_2\big]+\mathbb{E}^{\frac{1}{k}}_\{y\sim \nu\}\big[||g_\omega(y)||^k_2\big])$ is used. This particular choice of the regularization term facilitates the proofs to Lemma 2 and subsequently to Corollary 1.1 with details in Appendix D. In fact, we have also examined other types of regularization terms such as the $L_2$ norm of the output of $g(\cdot)$, and empirically they produce similar numerical results as the current regularization term."

---

> > ### Author Response · Authors · 2021-11-20
> > **Response to Reviewer RRsR (2/3)**
> >
> > **(2) Since the projection is made by the same mapping $g$, it might be not guaranteed maximal discrimination over the projected samples. Intuitively, I think the mapping $g$ should be data-dependent and/or making a trade-off with different regularization parameters, i.e. the functional $L(\mu, \nu, \lambda ; g)$ becomes $L(\mu, \nu, \lambda_{1}, \lambda_{2} ; g)$.**
> >
> > **[Since the projection is made by the same mapping $g$, it might be not guaranteed maximal discrimination over the projected samples.]}** Given two compared measures $\mu, \nu\in P_k(\mathbb{R}^d)$, if the projections are made by using different mappings $g_1$ and $g_2$ depending on $\mu$ and $\nu$, then the resulting distance can be difficult to interpret and may not reflect the dissimilarity between the compared measures, because the projections of samples from $\mu$ and $\nu$ will reside in different hypersurfaces, which may not be comparable. Besides, since it is not compatible with the framework of the SWD and the GSWD, the resulting distance may not be a valid metric. For instance, given a probability measure $\mu\in P_k(\mathbb{R}^d)$, the identity of indiscernibles does not hold with different mappings $g_1$ and $g_2$:
> >
> > $
> > \begin{equation}
> >     \text{ASWD}_\{k\}(\mu, \mu ; g_1, g_2)=\bigg(\int_\{\mathbb{S}^{d_\{\theta\}-1}\} W_\{k\}^{k}(\mathcal{H}_\{\mu\}(\cdot, \theta ; g_1), \mathcal{H}_\{\mu\}(\cdot, \theta ; g_2)) d \theta\bigg)^{\frac{1}{k}}\neq 0 .
> > \end{equation}
> > $
> >
> > **[Intuitively, I think the mapping $g$ should be data-dependent]** In the numerical implementation of the ASWD given in Section 3.2, the mapping $g$ is optimized with respect to the training data, thus the optimal parameter ($\omega$) for $g$ is data-dependent.
> >
> > **[Making a trade-off with different regularization parameters, i.e. the functional $L(\mu, \nu, \lambda ; g)$ becomes $L(\mu, \nu, \lambda_\{1\}, \lambda_\{2\} ; g)$.]**
> >
> > If there is a trade-off between different regularization terms, i.e. the functional $L(\mu, \nu, \lambda ; g)$ becomes $L(\mu, \nu, \lambda_\{1\}, \lambda_\{2\} ; g)$, then the optimal mapping $g^\{*\}(\cdot)$ defined by Equation (17) becomes:
> >
> > $
> > \begin{align}
> > g^{*}(\cdot)&\overset{\triangle}{=}\underset{g}{\operatorname{argmax}}\\{\operatorname{ASWD}_\{k\}(\mu, \nu ; g)-L(\mu, \nu, \lambda_1, \lambda_2 ; g)\\}, (1)
> > \end{align}
> > $
> >
> > where $L(\mu, \nu, \lambda_\{1\}, \lambda_\{2\} ; g)=\big(\lambda_1\mathbb{E}_\{x \sim \mu\}^{\frac{1}{k}}[\|g(x)\|_\{2\}^{k}]+\lambda_2\mathbb{E}_\{y \sim \nu\}^{\frac{1}{k}}[\|g(y)\|_\{2\}^{k}]\big)$.
> >
> > Consequently for $\lambda_1 \neq \lambda_2$, optimizing $\\{\operatorname{ASWD}_\{k\}(\mu, \nu ; g)-L(\mu, \nu, \lambda_1, \lambda_2 ; g)\\}$ and $\\{\operatorname{ASWD}_\{k\}(\nu, \mu ; g)-L(\nu, \mu, \lambda_1, \lambda_2 ; g)\\}$ will result in two different mappings $g^*_1(\cdot)$ and $g^*_2(\cdot)$, respectively, which implies the resulting distance is not symmetric and thus not a valid metric.
> >
> > **(3) As was shown in Remark 3 that if the tuning parameter $\lambda>1$ then ASWD is a proper metric, whereas the authors conducted their experiments with values $\lambda<1$, because it leads to outperforming the other sliced-based Wasserstein metrics. In my opinion, this restricts the applications of ASWD, since one has to investigate a panel of tuning parameters $\lambda<1$ to first check if ASWD is still a valid distance then if it gives good performance compared to the other SWDs.**
> >
> > As discussed in Remark 3, $\lambda >1$ is a sufficient but not necessary condition for the ASWD to be a metric. It is one sufficient condition to guarantee that the output of $g^*(x)$ to be finite for $\forall x\sim\mu,\nu$, which is required for Equation (49) to hold and ASWD to be a valid metric. In practice, $\lambda<1$ can also lead to finite output of $g^*(x)$ and subsequently valid metric as discussed in details in Remark 6. In all of experiments we conducted, including a wide range of values $\lambda \in \\{0.01, 0.05, 0.1, 0.5, 1.01, 10, 100\\}$, we found $g^*(x)$ to be finite which is expected with finite parameter values of neural networks. We would like to further clarify that only in the sliced Wasserstein flow experiment, we reported results with $\lambda<1$, while $\lambda>1$ is used in all the other experiments including the generative modelling experiment, the sliced Wasserstein auto-encoder experiment, etc. In these experiments, the ASWDs with $\lambda>1$ also produced better performance than the other slice-based Wasserstein metrics.

---

> > > ### Author Response · Authors · 2021-11-20
> > > **Response to Reviewer RRsR (3/3)**
> > >
> > > **(4) Does it possible to construct $g$ by other injective NNs?**
> > >
> > > Yes, it is possible to construct $g$ by other injective NNs. As discussed in Section 5.1 and Appendix G.2, we have investigated the effect of the choice of injective neural networks. Specifically, in Appendix G.2, we reported in Figure 9 the performance of the ASWD defined with other types of injective mappings different from that specified through Equation (18). In particular, two types of invertible mappings, the planar flow and the radial flow (Rezende & Mohamed, 2015), were evaluated. We found that although the ASWD defined with the planar flow and the radial flow has slightly worse performance than the ASWD defined with Equation (18), which we believe is due to the additional restriction imposed by invertible mappings, they still exhibit better performance than the other evaluated slice-based Wasserstein metrics.
> > >
> > > **(5) How is the dimensionality of the latent space $d_\theta$ decided and is its impact?**
> > >
> > > We have now investigated the impact of the dimensionality $d_\theta$ of the augmented space on the performance of the ASWDs and have reported the numerical results in Appendix G.2 of the revised paper draft.
> > >
> > > Specifically, the subsection of **"Choice of the dimensionality $d_\theta$ of the augmented space"** is added to Appendix G.2, which reports that "To investigate how the dimensionality $d_\theta$ of the augmented space affects the performance of the ASWD, different choices of $d_\theta$ are employed in the ASWD. Specifically, the injective network architecture $g_\omega(x)=[x, \phi_\omega(x)]: \mathbb{R}^d\rightarrow\mathbb{R}^{d_\theta}$ given in Equation (18) is adopted and $\phi_\omega$ is set to be single fully-connected neural networks whose output dimension equals \{1, 2, 3, 4\} times its input dimension, i.e. $d_\theta=\\{2d, 3d, 4d, 5d\\}$, respectively, where $d$ is the dimensionality of $x$. The numerical results are presented in Figure 10, and it can be found that the ASWDs present similar results across different choices of $d_\theta$. From Figure 10, the ASWDs with different choices of $d_\theta$ consistently produce better performance than the other evaluated slice-based Wasserstein metrics.''
> > >
> > > **(6) Type: page 4, (Definition 1) "Given an measurable" $\rightarrow$ "Given a measurable."**
> > >
> > > Thank you for pointing out the typo which we have fixed in the revised draft.
> > >
> > > **References**
> > >
> > > D. Rezende and S. Mohamed. Variational inference with normalizing flows. In *International Conference on Machine Learning (ICML)*, pp. 1530–1538, Lille, France, 2015.

---

> > > > ### Comment · Reviewer_RRsR · 2021-12-01
> > > > **After rebuttal**
> > > >
> > > > I thank the authors for their thorough response to my questions and concerns.

---

### Official Review · Reviewer_ViRm · 2021-11-10

**Correctness:** 4
**Technical Novelty And Significance:** 3
**Empirical Novelty And Significance:** 3
**Recommendation:** 6
**Confidence:** 3

**Main Review:**

Strengths
- The paper is well written and results are easy to follow.
- The new metric is well justified with respect to existing literature, and many of its theoretical properties are analyzed.
- The amount of numerical experiments and the variety of problems is very impressive!

Weaknesses
- Many of the points where mathematical formality is required is unfortunately omitted. For example at the beginning of page 4, I'm still wondering what are those "certain conditions". How large or small, or scalable is the approximation in (10) or (11)? What are the "certain non-trivial requirements" in Section 3.1. What are the "certain conditions" in Remark 1? What are the "certain constraints " in Section 3.2 Optimization objective?

**Summary Of The Paper:**

This manuscript introduces the concept of augmented sliced Wasserstein distances. The main idea is to extend the sliced Wasserstein distance based on mapping samples to higher -dimensional hypersurfaces. The proposed distance is shown to be a metric. Moreover, given that the optimal choice of the nonlinear maps is rather computationally intensive to obtain, an approximation based on neural networks is proposed. Several experiments are shown where a better performance is obtained with respect to existing methods.

**Summary Of The Review:**

Correctness:
- I believe the statements are correct, but I did not checked ever single line of all the proofs.

Novelty:
- Many of the concepts are natural extensions of the sliced Wasserstein distances. However, there is sufficient novelty.

Empirical novelty:
- A very complete numerical examples set is presented.

---

> ### Author Response · Authors · 2021-11-20
> **Response to Reviewer ViRm (1/2)**
>
> Many thanks for your positive and constructive feedback! Please see below our detailed response to your comments:
>
> **(1) At the beginning of page 4 , I'm still wondering what are those "certain conditions". What are the "certain non-trivial requirements" in Section 3.1?**
>
> As stated in (Kolouri et al., 2019) and (Beylkin, 1984), $\beta$ is a defining function of the GRT if it meets four conditions **H.1 - H.4** given in Section $3.1$ of (Kolouri et al., 2019). In addition, Theorem 1 and Corollary 1 of (Homan \& Zhou, 2017) provide sufficient conditions for GRTs to be bijective. To remove this ambiguity, we have added in the revised paper the conditions the defining function $\beta(\cdot)$ needs to satisfy to make the GRT bijective by adding the following text after Equation (5) of the paper:
>
> "In particular, a function $\beta(x, \theta)$ defined on $\mathcal{X} \times\left(\mathbb{R}^{d} \backslash\{0\}\right)$ with $\mathcal{X} \subseteq \mathbb{R}^{d}$ is called a defining function of GRTs if it satisfies conditions **H. 1 - H.4** given in (Kolouri et al., 2019)."
>
> Besides, we have also changed
>
> "Notably, the Radon transform is a linear bijection (Helgason, 1980), and the GRT is a bijection if the defining function $\beta$ satisfies certain conditions (Beylkin, 1984)."
>
> to
>
> "Notably, the Radon transform is a linear bijection (Helgason, 1980), and the sufficient conditions for GRTs to be bijective are provided in (Homan \& Zhou, 2017)."
> In Section 3.1, the "certain non-trivial requirements" are clarified by changing:
>
> "However, it is not straightforward to design defining functions $\beta(x, \theta)$ (Kolouri et al., 2019) for the GRT due to certain non-trivial requirements for the function (Beylkin, 1984)."
>
> to:
>
> "However, it is not straightforward to design defining functions $\beta(x, \theta)$ for the GRT, since one needs to first check if $\beta(x, \theta)$ satisfies the conditions to be a defining function (Kolouri et al., 2019; Beylkin, 1984), and then whether the corresponding GRT is bijective or not (Homan \& Zhou, 2017)."
>
> **(2) How large or small, or scalable is the approximation in (10) or (11)?**
>
> According to (Nadjahi et al., 2020), the overall complexity of the approximation error $\left|\widehat{\operatorname{SWD}}_{k, L}^k\left(\hat{\mu}_N, \hat{\nu}_N\right)-\operatorname{SWD}_k^{k}(\mu, \nu)\right|$ in estimating the SWD using Equations (10) is bounded by the sum of the sample complexity and the projection complexity, where
> $\widehat{\text{SWD}}_\{k, L\}^{k}(\hat{\mu}_N, \hat{\nu}_N)$ is an estimation of the SWD using $L$ projections between the empirical measures $\hat{\mu}_N, \hat{\nu}_N$ with $N$ samples. Specifically, for $k$-SWD and any $\mu, \nu \in P_q(\mathbb{R}^d)$ with $q>k$, and corresponding empirical measures $\hat{\mu}_N, \hat{\nu}_N$ with $N$ samples, (Nadjahi et al., 2020) derived the following upper bound for the sample complexity of SWD:
>
> $
> \begin{aligned}
> &\mathbb{E}\left|\text{SWD}_k\left(\hat{\mu}_N, \hat{\nu}_N\right)-\text{SWD}_k(\mu, \nu)\right| \leq C_\{k, q\}^{1 / k} M_\{q\}^{1 / q}(\mu, \nu) \begin{cases}N^{-1 /(2 k)} & \text { if } q>2 k\\\\
> N^{-1 /(2 k)} \log (N)^{1 / k} & \text { if } q=2 k\\\\
> N^{-(q-k) /(k q)} & \text { if } q \in(k, 2 k)\end{cases}, (1)
> \end{aligned}
> $
>
> where $M_q(\mu)$ refers to the moment of order $q$ of $\mu$ and $M_\{q\}^{1 / q}(\mu, \nu)=M_\{q\}^{1 / q}(\mu)+M_\{q\}^{1 / q}(\nu)$. Analyzing the sample complexity of the GSWD was mentioned in (Nadjahi et al., 2020) as a future research direction. The projection complexities of Equation (10) are also provided in (Nadjahi et al., 2020) and can be easily extended to that of Equation (11). In particular, the error caused by the Monte Carlo sampling with $L$ projections can be bounded as follows:
>
> $
> \begin{align}
> &\left\\{\mathbb{E}\left|\widehat{\text{SWD}}_\{k, L\}^{k}(\mu, \nu)-\text{SWD}_\{k\}^{k}(\mu, \nu)\right|\right\\}^{2}\\\\
> &\leq L^{-1} \int_\{\mathbb{S}^{d-1}\} \bigg\\{W_k^{k}\left(\mathcal{R}_\{\mu\}(\cdot, \theta), \mathcal{R}_\{\nu\}(\cdot, \theta)\right)-\bar{\delta}_\{k\} \bigg\\} \mathrm{d}{\sigma}(\theta), &(2)\\\\
> &\left\\{\mathbb{E}\left|\widehat{\text{GSWD}}_\{k, L\}^{k}(\mu, \nu)-\text{GSWD}_\{k\}^{k}(\mu, \nu)\right|\right\\}^{2}\\\\
> &\leq L^{-1} \int_\{\mathbb{S}^{d_\theta-1}\} \left\\{W_k^{k}\left(\mathcal{G}_\{\mu\}(\cdot, \theta), \mathcal{G}_\{\nu\}(\cdot, \theta)\right)-\bar{\zeta}_\{k\}\right\\} \mathrm{d}{\sigma}(\theta), &(3)\\\\
> \end{align}
> $
>
> where $\bar{\delta}_\{k\}=\int_\{\mathbb{S}^{d-1}\}W_k^{k}\left(\mathcal{R}_\{\mu\}(\cdot, \theta), \mathcal{R}_\{\nu\}(\cdot, \theta)\right)\mathrm{d}{\sigma}(\theta)$, $\bar{\zeta}_\{k\}=\int_\{\mathbb{S}^{d_\theta-1}\}W_k^{k}\left(\mathcal{G}_\{\mu\}(\cdot, \theta), \mathcal{G}_\{\nu\}(\cdot, \theta)\right)\mathrm{d}{\sigma}(\theta)$.

---

> > ### Author Response · Authors · 2021-11-20
> > **Response to Reviewer ViRm (2/2)**
> >
> > We have added text after Equations (10) and (11) to provide more details on their approximation errorand scalability:
> >
> > “The approximation error in estimating SWDs using Equation (10) is derived in (Nadjahi et al., 2020). ”.
> >
> > **(3) What are the "certain conditions" in Remark 1?**
> >
> > In Remark 1, to inherit the theoretical properties of the Radon transform, the function $g(\cdot)$ is required to be a measurable injective mapping $g(\cdot): \mathbb{R}^d \rightarrow \mathbb{R}^{d_\theta}$ as in Definition 1. We have removed this ambiguity in the revised paper by revising Remark 1 from:
> >
> > "Note that the spatial Radon transform can be interpreted as applying the vanilla Radon transform to $\hat{\mu}_g$, where $\hat{\mu}_g$ refers to the push-forward measure $g_\\#\mu$, i.e. the spatial Radon transform defined by Equation (13) can be rewritten as:
> >
> > $
> > 	\begin{align}
> > 	\mathcal{H}_\{\mu\}(t, \theta;g)&=E_\{x\sim \mu\}[\delta(t-\langle g(x), \theta \rangle)]\\\\
> > 	&=E_\{\hat{x}\sim \hat{\mu}_g\}[\delta(t-\langle \hat{x}, \theta \rangle)]\\\\
> > 	&=\int \delta(t-\langle \hat{x}, \theta \rangle)d\hat\{\mu\}_g\\\\
> > 	&=\mathcal{R}_\{\hat{\mu}_g\}(t, \theta).\\\\
> > 	\end{align}
> > $
> >
> > Hence the spatial Radon transform inherits the theoretical properties of the Radon transform subject to certain conditions of $g(\cdot)$ and incorporates nonlinear projections through $g(\cdot)$."
> >
> > to:
> >
> > "Note that the spatial Radon transform can be interpreted as applying the vanilla Radon transform to $\hat{\mu}_g$, where $\hat{\mu}_g$ refers to the push-forward measure $g_\\#\mu$, i.e. given a measurable injective mapping $g(\cdot): \mathbb{R}^d\rightarrow \mathbb{R}^{d_\theta}$, the spatial Radon transform defined by Equation (13) can be rewritten as:
> >
> > $
> > 	\begin{align}
> > 	\mathcal{H}_\{\mu\}(t, \theta;g)&=E_\{x\sim \mu\}[\delta(t-\langle g(x), \theta \rangle)]\\\\
> > 	&=E_\{\hat{x}\sim \hat{\mu}_g\}[\delta(t-\langle \hat{x}, \theta \rangle)]\\\\
> > 	&=\int \delta(t-\langle \hat{x}, \theta \rangle)d\hat\{\mu\}_g\\\\
> > 	&=\mathcal{R}_\{\hat{\mu}_g\}(t, \theta).\\\\
> > 	\end{align}
> > $
> >
> > Hence the spatial Radon transform inherits the theoretical properties of the Radon transform and incorporates nonlinear projections through $g(\cdot)$."
> >
> > **(4) What are the "certain constraints " in Section 3.2 optimization objective?**
> >
> > Here the "certain constraints" are implicitly realized by maximizing the objective given in Equation (19), which prevents the norm of the projected samples on the hypersurfaces from being arbitrarily large.
> >
> > We have clarified the "certain constraints" by changing:
> >
> > "We aim to slice distributions with maximally discriminating hypersurfaces between two distributions, so that the projected samples between distributions are most dissimilar subject to certain constraints on the hypersurface, as shown in Figure 1."
> >
> > to:
> >
> > "We aim to slice distributions with maximally discriminating hypersurfaces between two distributions while avoiding the projected samples being arbitrarily large, so that the projected samples between the compared distributions are finite and most dissimilar regarding the ASWD, as shown in Figure 1."
> >
> > **References**
> >
> > G. Beylkin. The inversion problem and applications of the generalized Radon transform. *Communications on Pure and Applied Mathematics*, 37(5):579-599, 1984.
> >
> > S. Helgason. *The Radon transform*, volume 2.  Basel, Switzerland: Springer, 1980.
> >
> > A. Homan and H. Zhou. Injectivity and stability for a generic class of generalized Radon transforms. *The Journal of Geometric Analysis*, 27(2):1515-1529, 2017.
> >
> > S. Kolouri, K. Nadjahi, U. Simsekli, R. Badeau, and G. Rohde. Generalized sliced Wasserstein distances. In *Proc. Advances in Neural Information Processing Systems (NeurIPS)*, pp. 261-272, Vancouver, Canada, 2019.
> >
> > K. Nadjahi et al. Statistical and topological properties of sliced probability divergences. In *Proc. Advances in Neural Information Processing Systems (NeurIPS)*, 33, 2020.

---

### Author Response · Authors · 2021-11-20
**Response to all reviewers**

We would like to thank all reviewers for their insightful suggestions to improve this paper and their appreciation of the paper and the proposed method, and we have reflect many valuable suggestions made by the reviewers in the revised paper. We would like to reiterate our key contributions before responding to each reviewer individually.

At a high level, this work proposes a new family of slice-based Wasserstein distance metrics with a novel incorporation of injective neural networks to learn nonlinear projections that can capture the complex structure of the data distributions. We address a major limitation of the state-of-the-art approaches in incorporating nonlinear projections into slice-based Wasserstein distances, i.e. the choice of defining functions are limited, problem specific, and is essentially a hyper-parameter that cannot be end-to-end optimized for constructing a valid distance metric. Compared with previous work, the ASWD is data-adaptive, can be end-to-end optimised, is a valid distance metric, and has been shown to lead to strong empirical performance in a wide range of benchmark experiments. We are thankful that our work has been appreciated by all reviewers as "The amount of numerical experiments and the variety of problems is very impressive!" (Reviewer ViRm), "The paper is well-written and the approach is mostly well presented" (Reviewer RRsR), "the contribution is of interest and significative."(Reviewer 1Q3z) and "well written and well motivated" (Reviewer 6LWm).

We summarize below the key revisions made in the revised paper draft according to each reviewers' comments. We are looking forward to having further discussions with the reviewers if there is anything that requires further clarification.

(1) (Reviewer ViRm) We have removed the referred mathematical ambiguity. Specifically, the "certain conditions" and "certain non-trivial requirements" in Section 3.1 are now given in the sentence after Equation (5) and Section 3.1. In addition, the "certain conditions" in Remark 1 has been explicitly provided, and the "certain constraints" in Section 3.2 has been rephrased to make the statement precise.

(2) (Reviewer RRsR) We have added in Appendix E the text explaining the particular choice of the regularization term used in the paper. The impact from the dimensionality $d_\theta$ of the augmented space is investigated in the subsection "choice of the dimensionality $d_\theta$ of the augmented space" of Appendix G.2.

(3) (Reviewer 1Q3z) We have studied in Appendix H.2 the effect of batch size on the computation cost of the ASWD and compared it to that of the SWD, and the DSWD. The impact from the dimensionality $d_\theta$ of the augmented space is investigated in the subsection "choice of the dimensionality $d_\theta$ of the augmented space" of Appendix G.2. We have also updated our implementation according to the latest repository of (Nguyen et al., 2021) and updated experiment results reported in Table 1, Figure 3, and Figure 11 of the paper. We have also rephrased Remark 4 to avoid confusion.

(4) (Reviewer 6LWm) We have added text in the subsection "Impact of the regularization coefficient" to study the effect of the regularization term on the performance of the Max-GSWD-NN.

**References**

K. Nguyen, N. Ho, T. Pham, and H. Bui. Distributional sliced-Wasserstein and applications to generative modeling. In *Proc. International Conference on Learning Representations (ICLR)*, Vienna, Austria, 2021.

---

### Decision · Program_Chairs · 2022-01-20

**Decision:**

Accept (Poster)

**Comment:**

The paper presents a variant of sliced wasserstein distance , where the slicing operation is performed with a neural network. The resulting distance is studied and experiments on synthetic data and as cost in generative modeling are performed.

While the idea of the paper is not that novel, the work is overall well executed. Reviewers agreed that the paper is borderline weak accept. Accept as a poster.